# Towards Multi-Spatiotemporal-Scale Generalized PDE Modeling

**Jayesh K. Gupta**[*]                                                           *jkg@cs.standford.edu*
*Microsoft Autonomous Systems and Robotics Research*

**Johannes Brandstetter**[*]                                                    *johannesb@microsoft.com*
*Microsoft Research AI4Science*

**Reviewed on OpenReview:** *https: // openreview. net/ forum? id= dPSTDbGtBY*

## Abstract

Partial differential equations (PDEs) are central to describing complex physical system simulations. Their expensive solution techniques have led to an increased interest in deep neural network based surrogates. However, the practical utility of training such surrogates is contingent on their ability to model complex multi-scale spatio-temporal phenomena. In recent years, various neural network architectures have been proposed to target such phenomena, most notably Fourier Neural Operators (FNOs), which give a natural handle over local & global spatial information via parameterization of different Fourier modes, and U-Nets which treat local and global information via downsampling and upsampling paths. However, large-scale comparisons between these convolution-based approaches are notoriously sparse. In this work, we make such comprehensive comparisons regarding performance, runtime complexity, memory requirements, and generalization capabilities. Concretely, we stress-test various FNO, (Dilated) ResNet, and U-Net like approaches to fluid mechanics problems in both vorticity-stream and velocity function form. For U-Nets, we transfer recent architectural improvements from computer vision, most notably from object segmentation and generative modeling. Next, we use our insights on design considerations, and introduce U-FNets, i.e., modern U-Nets that are augmented with FNO downsampling layers. Those architectures further improve performance without major degradation of computational cost. Finally, we ablate and discuss various choices for parameter conditioning, and show promising results on generalization to different PDE parameters and time-scales with a single surrogate model. Source code for our PyTorch benchmark framework is available at https://github.com/microsoft/pdearena.

## 1 Introduction

Many mathematical models of physical phenomena are expressed in differential equation forms (Olver, 1986), generally as temporal partial differential equations (PDEs). Their expensive solution techniques have led to an increased interest in deep neural network based surrogates (Bar-Sinai et al., 2019; Raissi et al., 2019; Lu et al., 2021; Li et al., 2020a; Brandstetter et al., 2022c; Um et al., 2020); especially in the studies that relate to fluid dynamics (Guo et al., 2016; Kochkov et al., 2021; Rasp & Thuerey, 2021; Keisler, 2022; Weyn et al., 2020; Sønderby et al., 2020; Wang et al., 2020a; Pathak et al., 2022). However, generalizing across different PDE parameters, and different time-scales is a notoriously hard problem. For example, in fluid mechanics, slightly different values of a single parameter like Reynolds numbers can make all the difference for flows being laminar or turbulent. Another source of challenge stems from the fact that physical phenomena appear at different spatial and temporal scales. For example, blizzards are rather local weather phenomena, whereas heat waves are rather global ones, both resulting from the same underlying principles. For exactly these

---

[*]Equal contributions.

reasons, fluid mechanics in general (Munson et al., 2013), and weather forecasting in particular (Jolliffe & Stephenson, 2012) have always posed a great scientific challenge.

Prominent examples of neural PDE surrogates are **Fourier Neural Operators (FNOs)** (Li et al., 2020a). At its core, FNO building blocks consist of Fast Fourier transforms (FFTs) (Cooley & Tukey, 1965; Van Loan, 1992) and weight multiplication in the Fourier space, where low Fourier modes provide global information, and high Fourier modes provide local information. An FNO layer processes global and local information simultaneously via weight multiplication of the different modes. On the other hand, **U-Nets** (Ronneberger et al., 2015) are standard architectures in the context of image modeling, image segmentation, and image generation. U-Nets are constructed

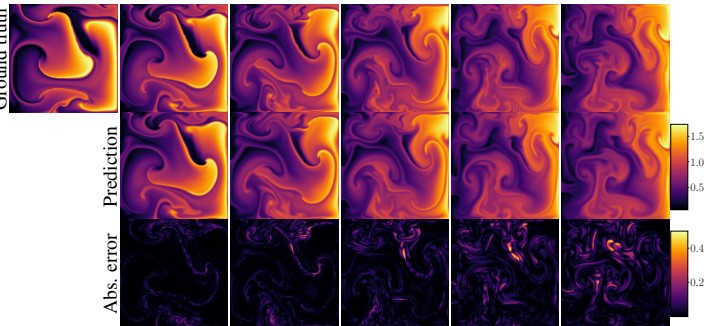

Figure 1: Example rollout trajectories of the best-performing U-Net model, which is trained to generalize across different timesteps ($\Delta t$) and different force terms.

as a spatial downsampling pass followed by a spatial upsampling pass, with additional skip connections present between the downsampling pass activations and corresponding upsampling layers. Local and global information is therefore treated in a more distributed fashion than in FNO like architectures. Downsampling corresponds to sequentially processing information more globally, whereas upsampling corresponds to fine-graining the global information and adding local information via skip connections. Figure 2 contrasts local and global information flows in FNO and U-Net like architectures. Given the recent success of modern U-Net architectures in complex generative image modeling tasks (Ho et al., 2020; Nichol & Dhariwal, 2021; Ramesh et al., 2021) it's pertinent that these are evaluated on PDE Operator learning tasks and compared to FNO like approaches. Furthermore, given the different nature of FNO and U-Net like approaches, it is worth surveying their respective advantages and performance on different tasks, as well as investigating under which circumstances combining them might be beneficial. The third line of models are **ResNet** (He et al., 2016a) like architectures, which a priori have no natural handle on processing local and global information – in contrast to recently introduced **Dilated ResNets** (Stachenfeld et al., 2021) which adapt filter sizes at different layers via dilated convolutions.

To summarize our contributions: (1) To our knowledge, we are the first to present a side-by-side analysis of FNO, (Dilated) ResNet, and U-Net like architectures on their ability to model complex multi-scale spatio-temporal phenomena. In doing so, we present new architecture designs based on modern updates to U-Nets. (2) We generalize to different PDE parameters and time-scales showing promising results for single surrogate models as exemplified in Figure 1. (3) We propose a unified PyTorch based framework for enabling easy side-by-side comparisons of various PDE operator learning methods which is available at https://github.com/microsoft/pdearena.

## 2 Preliminaries

**Common parameterization of Fourier transform layers.** The discrete Fourier transform (DFT) together with point-wise multiplication in the Fourier space is the heart of Fourier Neural Operator (FNO) layers. DFTs convert an $n$-dimensional complex signal $f(x) = f(x_1, \ldots, x_n) : \mathbb{R}^n \to \mathbb{C}$ at $M_1 \times \ldots \times M_n$ grid points into its complex Fourier modes $\hat{f}(\xi_1, \ldots, \xi_n)$ via:

$$\hat{f}(\xi_1, \ldots, \xi_n) = \mathcal{F}\{f\}(\xi_1, \ldots, \xi_n) = \sum_{m_1=0}^{M_1} \ldots \sum_{m_n=0}^{M_n} f(x) \cdot e^{-2\pi i \cdot \left( \frac{m_1 \xi_1}{M_1} + \ldots + \frac{m_n \xi_n}{M_n} \right)} \, , \tag{1}$$

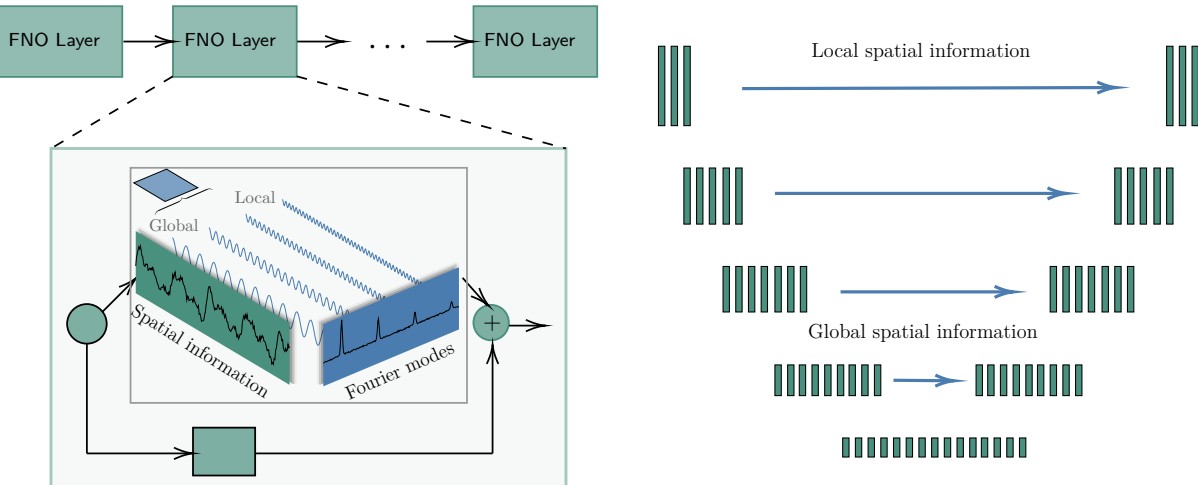

Figure 2: Information flow in Fourier based (left) and U-Net based architectures (right). FNO layers (Li et al., 2020a) consist of Fast Fourier transforms and weight multiplication in the Fourier space. Low Fourier modes provide global and high Fourier modes provide local information. U-Nets (Ronneberger et al., 2015) are constructed as a spatial downsampling pass, followed by a spatial upsampling pass, where information from the downsampling pass is added via skip-connections.

where $(\xi_1, \ldots, \xi_n) \in \mathbb{Z}_{M_1} \ldots \times \ldots \mathbb{Z}_{M_n}$. In FNO layers, discrete Fourier transforms on real-valued input fields and respective back-transforms – implemented as Fast Fourier Transforms[1] on real-valued inputs (RFFTs)[2] – are interleaved with a weight multiplication by a complex weight matrix of shape $c_{\text{in}} \times c_{\text{out}}$ for each mode, which results in a complex-valued weight tensor of the form $W \in \mathbb{C}^{c_{\text{in}} \times c_{\text{out}} \times (\xi_1^{\max} \times \ldots \times \xi_n^{\max})}$, where Fourier modes above cut-off frequencies $(\xi_1^{\max}, \ldots, \xi_n^{\max})$ are set to zero. These cut-off frequencies turn out to be important hyperparameters. Additionally, a residual connection is usually implemented as convolution layer with kernel size 1 (see Figure 2).

**Common parameterizations of convolution layers.** Regular convolutional neural network (CNN) (Fukushima & Miyake, 1982; LeCun et al., 1998) layers are the basic building blocks of U-Net like architectures. CNNs take as input feature maps $f : \mathbb{Z}^n \to \mathbb{R}^{c_{\text{in}}}$ and convolve[3] them with a set of $c_{\text{out}}$ filters $\{w^i\}_{i=1}^{c_{\text{out}}}$ with $w^i : \mathbb{Z}^n \to \mathbb{R}^{c_{\text{in}}}$:

$$[f \star w^i](x) = \sum_{y \in \mathbb{Z}^n} \langle f(y), w^i(y - x) \rangle , \tag{2}$$

which can be interpreted as an inner product of input feature maps with the corresponding filters at every point $y \in \mathbb{Z}^n$. The filter size of a convolutional layer is a crucial choice in neural network design since it defines the regions from which information is obtained. Common practice is to use rather small filters (Simonyan & Zisserman, 2014; Szegedy et al., 2015; He et al., 2016a). Continuous formulations of filters were introduced to handle irregularly sampled data (Schütt et al., 2018; Simonovsky & Komodakis, 2017; Wang et al., 2018; Wu et al., 2019) and to match the resolution of the underlying data (Peng et al., 2017; Cordonnier et al., 2019; Romero et al., 2021b). A promising direction is to adapt filter sizes at different layers using dilation (Dai et al., 2017). For example, dilated convolutions in the context of PDE modeling were proposed in Stachenfeld et al. (2021). As a downside, dilations might limit the bandwidth of the filters, and thus the amount of collected detail. A rather new direction is therefore to adapt filter sizes either via learnable dilation (Pintea et al., 2021) or via flexible sized continuous convolutions (Romero et al., 2021a; 2022).

---

[1]Fast Fourier transforms (FFTs) immensely accelerate DFT computation by factorizing Equation 1 into a product of sparse (mostly zero) factors.

[2]The FFT of a real-valued signal is Hermitian-symmetric, so the output contains only the positive frequencies below the Nyquist frequency for the last spatial dimension.

[3]In deep learning, a convolution operation in the forward pass is implemented as cross-correlation.

Figure 3: Analyzing filter properties of trained U-Net architectures. Absolute values of Fourier modes of the filters in each first layer of the respective down-sampling blocks are shown, where for each mode the average is taken over all filters.

**Connecting Fourier transform and convolution.** Starting with the 1-dimensional case and omitting channel dimensions, we assume a signal consisting of $n$ input points, and we further assume circular padding. We can now rewrite Equation 2 into a discrete *circular convolution* (Bamieh, 2018; Bronstein et al., 2021) of two $n-$dimensional vectors $\boldsymbol{f}, \boldsymbol{w} \in \mathbb{R}^n$:

$$[\boldsymbol{f} \star \boldsymbol{w}]_i = \sum_{j=0}^{n} \boldsymbol{w}_{(i-j) \bmod n} \boldsymbol{f}_j = \sum_{j}^{n-1} (\mathbf{C}_w)_{ij} \boldsymbol{f}_j \ , \quad \mathbf{C}_w = \begin{pmatrix} w_1 & w_2 & \dots & w_{n-1} & w_0 \\ w_0 & w_1 & w_2 & \dots & w_{n-1} \\ \vdots & & & \ddots & \vdots \\ w_2 & \dots & w_{n-1} & w_0 & w_1 \end{pmatrix} \ . \quad (3)$$

The indexing $(i-j) \bmod n$ returns circular shifts, which can be combined into a circulant matrix $\mathbf{C}_w$. It is general practice to use rather small filters which only consist of $k$ non-zero elements where usually $k << n$. The remaining elements of $\mathbf{C}_w$ are filled up with zeros. The action of $\mathbf{C}_w$ on $\boldsymbol{f}$, or equivalently the convolution of $\boldsymbol{f}$ with $\boldsymbol{w}$ can be expressed via the convolution theorem:

$$\mathbf{C}_w \boldsymbol{f} = \left( \frac{1}{\sqrt{n}} \mathbf{W} \right) \mathbf{D} \left( \frac{1}{\sqrt{n}} \mathbf{W}^* \right) \boldsymbol{f} \ , \quad (4)$$

where the matrix $\mathbf{W}$ consists of the eigenvectors of $\mathbf{C}_w$ and $\mathbf{W}^*$ is its complex conjugate. All circulant matrices have the same eigenvectors, which if multiplied with a signal yields the discrete Fourier transform (DFT) of the signal. That is, multiplication (from left) with $\mathbf{W}^*$ is the discrete Fourier transform (of $\boldsymbol{f}$), and multiplication by $\mathbf{W}$ is the inverse Fourier transform. The matrix $\mathbf{D}$ has the Fourier modes of the vector $\boldsymbol{w}$ on its diagonal. Thus, we can analyze a convolution by expressing its filters as vectors $\boldsymbol{w} \in \mathbb{R}^n$, which comprise the actual $k$ filter values ($k$ corresponds to the kernel size) and additional $n-k$ zeros. When we extend the circular convolution approach to two dimensions, $\boldsymbol{w}$ itself becomes a matrix $\boldsymbol{w} \in \mathbb{R}^{n \times n}$. In Figure 3, we plot the Fourier modes of the two dimensional filters of a trained U-Net. We take the absolute values of the modes and average for each mode over all filters in the first convolution layer of different down-sampling blocks. Although precise statements are difficult to make, it is evident that Fourier mode averages of different blocks are downsampled versions of each other, which complies with the interpretation that the downsampling blocks of U-Nets process information at different scales. This is therefore in contrast to FNO like architectures which process different scales within each FNO layer.

**Fourier transform for downsampling.** Bandlimited pre-subsampling (Mallat, 1999), i.e., suppressing high-frequencies before down-sampling, is a well know technique in signal processing; for an illustrative example see e.g., Figure 1 in Worrall & Welling (2019). We hypothesize that replacing convolutions with FNO layers which set Fourier modes above cut-off frequencies to zero is advantageous, especially in the lower downsampling blocks of U-Net architectures where the spatial resolution is large. In the lower U-Net blocks, convolutions might be all what is needed to learn efficient coarse-grained representations.

**Tested partial differential equations.** A partial differential equation (PDE) relates solutions $\boldsymbol{u} : \mathcal{X} \rightarrow \mathbb{R}^n$ and respective derivatives for all points $\boldsymbol{x}$ in the domain $\mathcal{X} \in \mathbb{R}^m$, where $\boldsymbol{u}^0(\boldsymbol{x})$ are *initial conditions* at time $t = 0$ and $B[\boldsymbol{u}](t, \boldsymbol{x}) = 0$ are *boundary conditions* with boundary operator $B$ when $\boldsymbol{x}$ is on the boundary $\partial \mathbb{X}$ of the domain. In this work, we investigate PDEs of fluid mechanics problems. To be more precise, we focus on the **incompressible Navier-Stokes** equations (Temam, 2001), and the shallow water

equations(Vreugdenhil, 1994) as an important variant thereof. For the latter, we test for data obtained in velocity function and vorticity stream formulation. Such comparisons, although never done for deep learning PDE surrogates, are important since the characteristics of these two formulations are different. While vorticity stream formulation is often easier to solve, it has limited applicability in complex and realistic flow scenarios, especially when dealing with complex geometries and resultant boundary conditions. Comparatively, velocity function formulation is more intuitive, directly dealing with physical measurable quantities, i.e., velocity fields, with easier implementation of boundary conditions (Gatski, 1991; Kundu et al., 2015).

In 2 dimensions, the Navier-Stokes equations in **vector velocity form** conserve the velocity flow fields $\boldsymbol{v} : \mathcal{X} \to \mathbb{R}^2$ where $\mathcal{X} \in \mathbb{R}^2$ via:

$$\frac{\partial \boldsymbol{v}}{\partial t} = -\boldsymbol{v} \cdot \nabla \boldsymbol{v} + \mu \nabla^2 \boldsymbol{v} - \nabla p + \boldsymbol{f} \ , \qquad \nabla \cdot \boldsymbol{v} = 0 \ , \tag{5}$$

where $\boldsymbol{v} \cdot \nabla \boldsymbol{v}$ is the convection, i.e. the rate of change of $\boldsymbol{v}$ along $\boldsymbol{v}$, $\mu \nabla^2 \boldsymbol{v}$ the viscosity, i.e. the diffusion or net movement of $\boldsymbol{v}$, $\nabla p$ the internal pressure and $\boldsymbol{f}$ an external force. An additional incompressibility constraint $\nabla \cdot \boldsymbol{v} = 0$ yields mass conservation of the Navier-Stokes equations.

By introducing the vorticity $\omega : \mathcal{X} \in \mathbb{R}$ as the curl of the flow velocity, i.e. $\omega = \nabla \times \boldsymbol{v}$, we can rewrite the incompressible 2-dimensional Navier-Stokes equations in **scalar vorticity stream function form** (Kundu et al., 2015; Guyon et al., 2001; Acheson, 1991) as:

$$\frac{\partial \omega}{\partial t} + \frac{\partial \psi}{\partial y} \frac{\partial \omega}{\partial x} + \frac{\partial \psi}{\partial x} \frac{\partial \omega}{\partial y} = \frac{1}{\mathrm{Re}} \left( \frac{\partial^2 \omega}{\partial x^2} + \frac{\partial^2 \omega}{\partial x^2} \right) \ , \qquad \left( \frac{\partial^2 \psi}{\partial x^2} + \frac{\partial^2 \psi}{\partial x^2} \right) = -\omega \ , \tag{6}$$

where the streamfunction is defined via the relations $\frac{\partial \psi}{\partial y} = \boldsymbol{v}_x$ and $\frac{\partial \psi}{\partial x} = -\boldsymbol{v}_y$, and Re is the Reynolds number which is indirectly proportional to the viscosity and proportional to the absolute velocity. As a result, the 2D incompressible Navier-Stokes equations are turned into one parabolic equation, i.e. the vorticity transport equation (Equation 6 left), and one elliptic equation, i.e. the Poisson equation (Equation 6 right). Since the streamfunction is directly obtained from the vorticity via the Poisson equation one usually solves for the scalar vorticity.

The **shallow water** equations (Vreugdenhil, 1994) can be derived from integrating the incompressible Navier–Stokes equations, in cases where the horizontal length scale is much larger than the vertical length scale. As such, shallow water equations describe a thin layer of fluid of constant density in hydrostatic balance, bounded from below by the bottom topography and from above by a free surface. For simplified weather modeling, the shallow water equations express the velocity in $x$- direction termed zonal velocity, the velocity in the $y$- direction termed meridional velocity, and the vertical displacement of free surface, which subsequently is used to derive pressure fields. Since the shallow water equations are derived from the Navier-Stokes equations, a vorticity-stream function formulation exists as well. Note however that when it comes to describing flows in e.g. more complex geometries, the velocity formulation is in general easier to deal with (Kundu et al., 2015).

## 3  PDE Surrogates

**(Dilated) ResNets.** We implement ResNet architectures using 8 residual blocks, where each block consists of two convolution layers with $3 \times 3$ kernels, shortcut connections, group normalization (Wu & He, 2018), and GeLU activation functions (Hendrycks & Gimpel, 2016). In contrast to standard ResNets for image classification, we don't use any down-projection techniques, e.g. convolution layers with strides larger than 1 or via pooling layers. In doing so, ResNets have no natural built-in handle over local and global informations, and therefore serve as important baseline to ablate effects of local and global information flow which is fundamental in e.g. FNO and U-Net like architectures. Recently, Stachenfeld et al. (2021) introduced Dilated ResNets, which adapt filter sizes at different layers using dilated convolutions, and thus are an alternative way of subsequently aggregating global information. The models consist of 4 residual blocks where each block individually consists of 7 dilated CNN layers with dilation rates of $[1, 2, 4, 8, 4, 2, 1]$. We implement Dilated ResNets with and without group normalization layers.

**Fourier Neural Operators.** We implement FNO architectures where the number of FNO layers, the number of channels, and the number of non-zero Fourier modes are hyperparameters. All architectures consist of two embedding and two output layers. Each FNO layer comprises a convolution path with a $1 \times 1$ kernel and a Fourier path where pointwise weight multiplication is done for the lower modes in the Fourier domain. We use GeLU activation functions, and no normalization scheme.

**U-Nets.** U-Nets have already been used as PDE surrogates in Ma et al. (2021); Chen & Thuerey (2021). U-Nets are constructed as a spatial downsampling followed by a spatial upsampling pass, where each down- and upsampling block consists of two convolutional layers. A particularity of U-Nets is the presence of skip connections between the downsampling pass activations and corresponding upsampling layers. Orignially, downsampling is achieved via max-pooling operations. We term the 2015 U-Net implementation as U-Net$_{2015}$, which is based on the `PDEbench` repository of Takamoto et al. (2022). Furthermore, we include a slightly different version which we term U-Net$_{base}$ which has bias weights and group normalization instead of batch normalization to be comparable with modern U-Net versions. To match the number of weights of U-Net$_{2015}$, the bottleneck layer in U-Net$_{base}$ is omitted. Modern versions of the architecture (Ho et al., 2020; Nichol & Dhariwal, 2021; Ramesh et al., 2021) often use Wide ResNet (Zagoruyko & Komodakis, 2016) style 2D convolutional blocks, each of which can be followed by a spatial attention block (Vaswani et al., 2017). Other notable changes are the substitution of max-pooling operations by downsampling layers. We term the respective implementations U-Net$_{mod}$ and U-Net$_{att}$ in our experiments.

**Fourier U-Nets.** Based on the insights of Section 2, we replace lower blocks both in the downsampling and in the upsampling path of U-Net architectures by Fourier blocks, where each block consists of 2 FNO layers and residual connections. We test substituting only the lowest block (U-F1Net), and the lowest two blocks (U-F2Net) of the U-Net$_{mod}$ architecture. Substituting all blocks would yield an architecture which resembles the UNO architecture (Rahman et al., 2022b), with the difference that in UNO architectures downsampling is done individually via linear layers along the $x$- and $y$- dimension, and that "mode scheduling" reduces the number of modes for higher blocks in the respective downsampling and upsampling paths. For complete comparison, we therefore also implement the UNO architecture[4].

## 3.1 Operator learning

Major practical benefits of neural PDE surrogates come from amortizing the cost of their compute-expensive training process which depends on the surrogates' ability to effectively generalize across different parameter settings as well as across different time discretizations. Operator learning is a popular term for training these neural surrogates. Theoretical grounding arises from Chen & Chen (1995) who extend the universal approximation theorem in neural networks (Hornik et al., 1989; Cybenko, 1989) to operator approximation, forming the basis for DeepONets (Lu et al., 2019) with theoretical extensions in Lu et al. (2021), graph kernel networks (Li et al., 2020b), and FNOs. An impressive comparison of DeepONets and FNOs can be found in Lu et al. (2022).

Operator learning (Lu et al., 2019; Li et al., 2020b;a; Lu et al., 2021; 2022) relates solutions $\boldsymbol{u} : \mathcal{X} \to \mathbb{R}^n$, $\boldsymbol{u}' : \mathcal{X}' \to \mathbb{R}^{n'}$ defined on different domains $\mathcal{X} \in \mathbb{R}^m$, $\mathcal{X}' \in \mathbb{R}^{m'}$ via operators $\mathcal{G}$:

$$\mathcal{G} : (\boldsymbol{u} \in \mathcal{U}) \to (\boldsymbol{u}' \in \mathcal{U}') , \tag{7}$$

where $\mathcal{U}$ and $\mathcal{U}'$ are the spaces of solutions $\boldsymbol{u}$ and $\boldsymbol{u}'$, respectively.

**Parameter conditioning.** We evaluate FNO and U-Net like architectures on their generalization capabilities across PDE parameters and different time-scales. Data points consist of solution pairs $\boldsymbol{u}, \boldsymbol{u}' \in \mathcal{U}$ where the pair itself is from the same solution space $\mathcal{U}$, but different pairs $\{\boldsymbol{u}, \boldsymbol{u}'\}_1$ and $\{\boldsymbol{u}, \boldsymbol{u}'\}_2$ are from different solution spaces $\mathcal{U}_1$ and $\mathcal{U}_2$ characterized by different PDE parameters, e.g., force terms. Simply put, an input-output pair is always drawn from one trajectory and therefore naturally shares the same PDE parameters. Further, the mapping $\boldsymbol{u} \to \boldsymbol{u}'$ should generalize across different time windows $\Delta t$, which is taken into account by conditioning on the lead time, i.e., the time the neural surrogate is asked to predict into the future. Taking all these requirements into account, we train neural surrogates to generalize across different

---

[4]We based our implementation on `https://github.com/ashiq24/UNO`

initial conditions, different PDE parameters (force terms) and different time windows. Both time windows $\Delta t$ and force terms are continuous scalar parameters, and thus can be encoded into a vector representation by using sinusoidal embeddings as is common in Transformers (Vaswani et al., 2017) and various neural implicit representation learning techniques (Mildenhall et al., 2021).

## 4  Experiments

We establish the following set of desiderata for our benchmarks: (i) *simplicity*: the tasks should be easy to setup, while being backed by actual PDE solvers written by domain experts, (ii) *challenging*: the tasks should be difficult enough, (iii) *diverse*: the tasks should be diverse, both in their formulation as well as in their requirements, and (iv) *generalizability*: the tasks should probe generalization across different time horizons as well as different parameter settings. Following these desiderata, we assessed the described architectures in four experimental settings to probe (i) Fourier vs. U-Net based approaches, (ii) differences due to the voriticty-stream vs. velocity function formulation of the datasets, and (iii) parameter conditioning performance. Results of the main paper are complemented by comprehensive studies and various ablations in Appendix B.

Concretely, we first probe selected architectures on the shallow water equations, both in vorticity-stream and velocity function form. We do a second comprehensive evaluation on the Navier-Stokes equations in velocity function formulation since this formulation turns out to be harder to model for neural surrogates. Finally, we probe parameter conditioning choices on the Navier-Stokes equations. We discuss results and differences between architectures.

All datasets contained multiple input and output fields. More precisely, one scalar and one velocity vector field in case of the velocity formulation, and two scalars in case of the vorticity formulation. Inputs to the neural PDE surrogates were respective fields at previous $t$ timesteps, where $t$ varies for different PDEs. The *one-step loss* is the mean-squared error at the next timestep summed over fields. The *rollout loss* (reported in Appendix B) is the mean-squared error after applying the neural PDE surrogate 5 times, summing over fields and time dimension. We alternatively test the relative MSE loss as used in Li et al. (2020a). We optimized models using the AdamW optimizer (Kingma & Ba, 2014; Loshchilov & Hutter, 2019) for 50 epochs and minimized the summed mean squared error. We used cosine annealing as learning rate scheduler (Loshchilov & Hutter, 2016) with a linear warmup. Table 1 compares parameter count, runtime and memory requirement of the tested architectures, showing that runtime and memory requirements are in the same ballpark for both architecture families if the number of parameters is kept similar.

**Shallow water equations.**  We modified the implementation in `SpeedyWeather.jl`[5](Klöwer et al., 2022), obtaining data on a grid with spatial resolution of $192 \times 96$ ($\Delta x = 1.875°$, $\Delta y = 3.75°$), and temporal resolution of $\Delta t = 48\,\mathrm{h}$. We first evaluated the different architectures on the shallow water equations in velocity function formulation, predicting scalar pressure field and vector wind velocity field. Figure 4 (left) shows results obtained by various models. In general, all methods which have a dedicated local and global information flow, i.e. Dilated ResNet, FNO, and U-Net architectures, perform rather well. Nevertheless, across all tested models, performance differences of an order of magnitude arise, where U-Nets in general perform best. Adding FNO blocks to U-Net architectures (U-F1Net, U-F2Net) seems to be beneficial. We further evaluate on the shallow water equations in vorticity stream formulation, and predict the scalar pressure field and the scalar wind vorticity field. Figure 4 (middle) shows results obtained by various models. Performance-wise a similar pattern arises, where again the lowest losses are observed for U-Net architectures.

**Velocity function formulation of Navier-Stokes equations.**  We further tested on Navier-Stokes equations in velocity function form, which is more common in the real world than the vorticity stream function form. In addition to the velocity field $\boldsymbol{v}$ of Equation 5, we introduced a scalar field representing a scalar quantity, i.e. particle concentration, that is being transported via the velocity field. The scalar field is *advected* by the vector field, i.e. as the vector field changes, the scalar field is transported along with it. Complementary, the scalar field influences the vector field only via an external buoyancy force term in $y$-direction, i.e. $\boldsymbol{f} = (0, f)^T$. We obtained data on a grid with spatial resolution of $128 \times 128$ ($\Delta x = 0.25$,

---

[5]https://github.com/milankl/SpeedyWeather.jl

Table 1: Comparison of parameter count, runtime, and memory requirement of various architectures. Subscript numbers indicate the used number of Fourier modes. For U-FNet experiments subscript numbers indicate the number of Fourier modes in the lowest and second-lowest block.

| METHOD | Channels | Res.Layers/Blocks | Params. | Runtime [s] | | Mem. [MB] | |
|---|---|---|---|---|---|---|---|
| | | | | Fwd. | Fwd.+bwd. | f32 size | Peak usage |
| ResNet128 | 128 | 8 | 2.4 M | 0.084 | 0.180 | 9 | 4273 |
| ResNet256 | 256 | 8 | 9.6 M | 0.231 | 0.497 | 38 | 8500 |
| DilResNet128 | 128 | 4 | 4.2 M | 0.118 | 0.342 | 16 | 4849 |
| DilResNet128-norm | 128 | 4 | 4.2 M | 0.183 | 0.423 | 16 | 6922 |
| FNO128-8$_{modes8}$ | 128 | 8 | 33.7 M | 0.057 | 0.162 | 134 | 2161 |
| FNO128-8$_{modes16}$ | 128 | 8 | 134 M | 0.059 | 0.171 | 537 | 2953 |
| FNO128-4$_{modes16}$ | 128 | 4 | 67.2 M | 0.031 | 0.089 | 268 | 1852 |
| FNO64-4$_{modes32}$ | 64 | 4 | 67.1 M | 0.016 | 0.050 | 268 | 1204 |
| FNO96-4$_{modes32}$ | 96 | 4 | 151 M | 0.026 | 0.080 | 604 | 2179 |
| FNO128-4$_{modes32}$ | 128 | 4 | 268 M | 0.036 | 0.118 | 1100 | 3420 |
| UNO64 | 64 | 7 | 110 M | 0.070 | 0.134 | 440 | 1925 |
| UNO128 | 128 | 7 | 440 M | 0.160 | 0.341 | 1800 | 5513 |
| U-Net$_{2015}$64 | 64 | 9 | 31 M | 0.013 | 0.037 | 124 | 1305 |
| U-Net$_{2015}$128 | 128 | 9 | 124 M | 0.042 | 0.117 | 496 | 3002 |
| U-Net$_{base}$64 | 64 | 8 | 31.1 M | 0.021 | 0.046 | 124 | 1277 |
| U-Net$_{base}$128 | 128 | 8 | 124 M | 0.056 | 0.132 | 496 | 3000 |
| U-Net$_{mod}$64 | 64 | 9 | 144 M | 0.079 | 0.184 | 577 | 3900 |
| U-Net$_{att}$64 | 64 | 9 | 148 M | 0.081 | 0.190 | 593 | 3975 |
| U-F1Net$_{modes8}$ | 64 | 9 | 154 M | 0.083 | 0.205 | 617 | 3936 |
| U-F1Net$_{modes16}$ | 64 | 9 | 185 M | 0.084 | 0.208 | 743 | 4037 |
| U-F2Net$_{modes8,4}$ | 64 | 9 | 163 M | 0.085 | 0.213 | 652 | 3961 |
| U-F2Net$_{modes8,8}$ | 64 | 9 | 193 M | 0.085 | 0.216 | 772 | 4046 |
| U-F2Net$_{modes16,8}$ | 64 | 9 | 224 M | 0.086 | 0.219 | 897 | 4149 |
| U-F2Net$_{modes16,16}$ | 64 | 9 | 344 M | 0.090 | 0.232 | 1400 | 4496 |

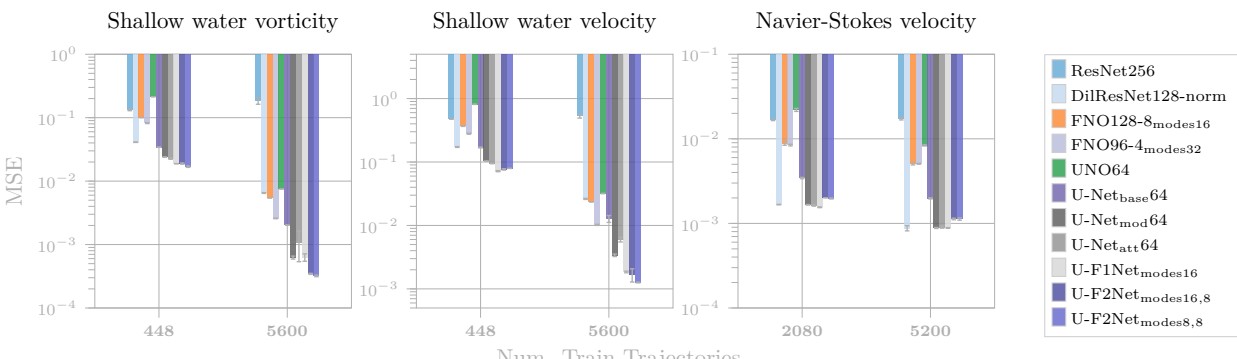

Figure 4: One-step errors for modeling different PDEs, shown for different number of training trajectories. Results are averaged over three different random seeds, and are obtained for the velocity function and vorticity stream formulation of the shallow water equations on 2-day prediction (left, middle), and for the Navier-Stokes equation (right). For better visibility only selected architectures are displayed, for full comparisons see Appendix B. Note the logarithmic scale of the $y$-axes.

$\Delta y = 0.25$), and temporal resolution of $\Delta t = 1.5$ s using $\Phi$Flow[6] (Holl et al., 2020). Figure 4 (right) shows results obtained by different architectures. In contrast to the shallow water experiments, the compute-expensive Dilated ResNet architectures perform on par with the best U-Net and U-FNet architectures. The reason for the performance difference of Dilated ResNets is hard to pin down. Our primary guess is on the importance of aliasing vs. patterns in the data. Concretely, for data with different proportions of higher frequency modes, the importance of aliasing might shift.

---

[6]https://github.com/tum-pbs/PhiFlow

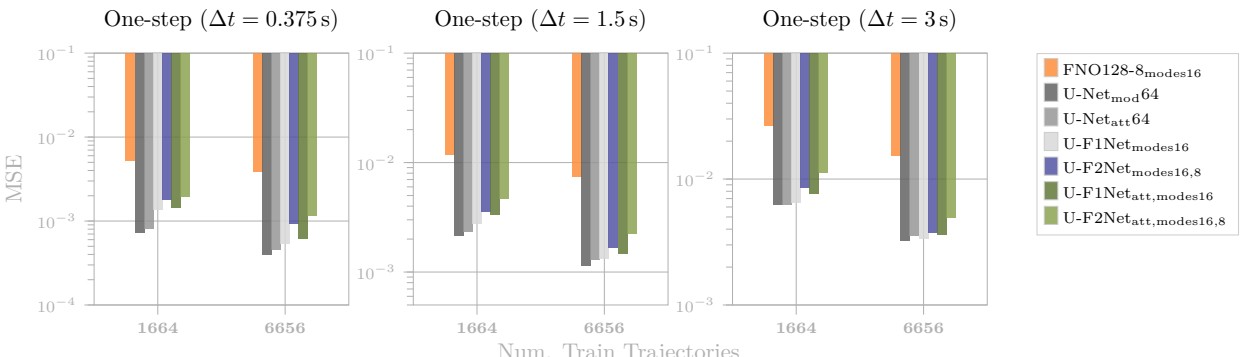

Figure 5: One-step errors obtained on the parameter conditioning experiments of the Navier-Stokes equation. Results are shown for selected architectures, different number of training trajectories, and different time windows: $\Delta t = 0.375\,\mathrm{s}$ (left), $\Delta t = 1.5\,\mathrm{s}$ (middle), and $\Delta t = 3\,\mathrm{s}$ (right). Results are averaged over 208 different unseen evaluation buoyancy force values between 0.2 and 0.5.

**Probing parameter conditioning.** We probe parameter conditioning on the velocity function formulation of the Navier-Stokes equation. We test FNO and U-(F)Net variants, experiments for Dilated ResNet are too compute-expensive due to their long runtimes, see Table 1. For training, we used a dataset with higher temporal resolution of $\Delta t = 0.375\,\mathrm{s}$ and get equal number of trajectories from uniformly sampling 832 different external buoyancy force values, $\boldsymbol{f} = (0, f)^T$ in Equation 5, in the range $0.2 \leq f \leq 0.5$, using input fields at one timestep.

We conditioned our models to predict for different time windows in the range $0.375\,\mathrm{s} \leq \Delta t \leq 20\,\mathrm{s}$, and different strengths of the $y$-component of the external buoyancy force $f$. Due to the unbalanced nature of the dataset size at different $\Delta t$, we reweighed the sampling frequency in our dataloader to try to maintain parity. We provided conditioning information in the form of an embedding vector which can be added to each or subset of residual blocks (Ho et al., 2020). Both, $\Delta t$ and $f$, are continuous valued scalar parameters, and thus can be encoded into a vector representation by using sinusoidal embeddings as is common in Transformers (Vaswani et al., 2017). We added the embedding vector to the feature maps after the first convolution/FNO layer in respective down- and up-sampling blocks. To be more precise, for each feature map we replicated the respective embedding value along $x$- and $y$-coordinates. For Fourier layers, this results in adding the embedding vector, the Fourier branch, and the residual connection together. We also compare an alternative conditioning approach for U-Nets termed AdaGN (Nichol & Dhariwal, 2021), based on affine transformation of group normalization layers via projections of our embeddings in Figure 19 of Appendix B.5.

Figure 5 shows results obtained by various models averaged over 208 different unseen evaluation force values. U-Net based methods perform best. In contrast to the unconditioned experiments, substituting lower blocks by FNO blocks didn't yield better generalization capabilities. In general, we observe that conditioning is more difficult for FNO layers, most strikingly seen in the performance curves of FNOs. We however do not discard the possibility that for FNO layers, alternative parameter embedding and conditioning methods might be required. Nevertheless, our results also coincide with the findings of Lu et al. (2022), which state that FNO like architectures seemed to be extremely sensitive to noise, and failed to predict solutions for even small amounts of added Gaussian noise.

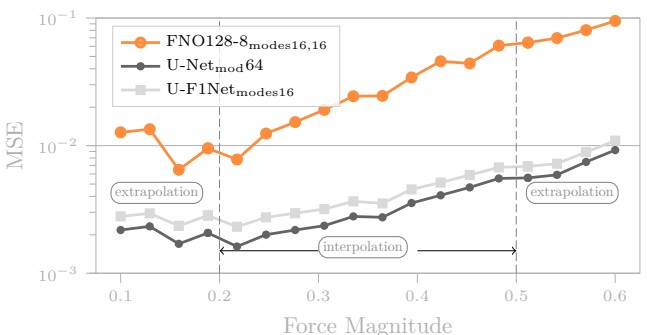

Figure 6: Inter- and extrapolation performance of different models tested on buoyancy force values in the range $0.1 \leq f \leq 0.6$ performing 5 steps rollout at $\Delta t = 0.375\,\mathrm{s}$.

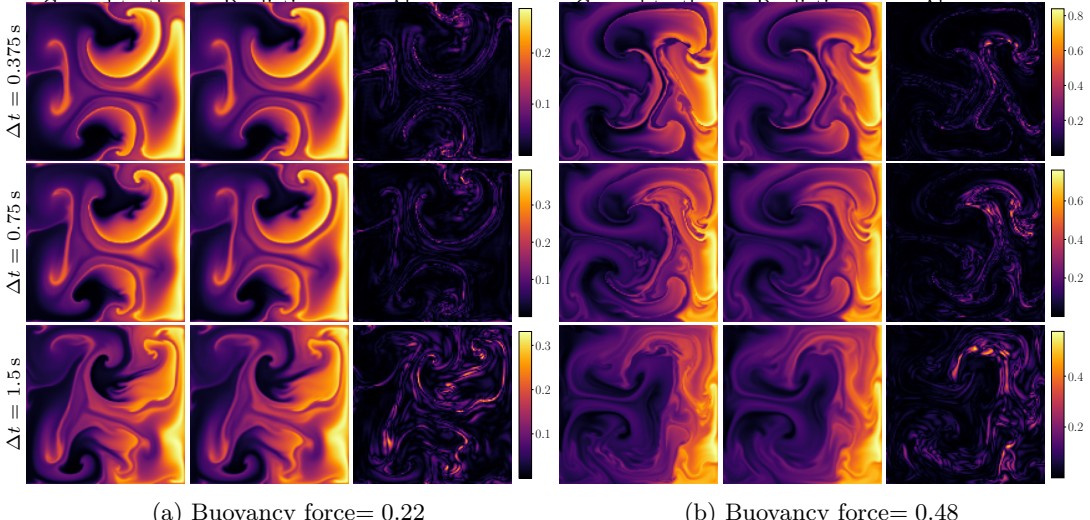

(a) Buoyancy force= 0.22            (b) Buoyancy force= 0.48

Figure 7: Scalar field predictions obtained for the parameter conditioning experiments using the best performing U-Net$_{\mathrm{mod}}$ model. Predicted and ground truth fields are shown for different buoyancy force values and different time windows. Model inputs are the same for different time window prediction tasks. More conditioning experiments can be found in Appendix B.5.

In Figure 6, we show performance of different models tested on buoyancy force values in the range $0.1 \leq f \leq 0.6$. The curves indicate that the difficulty of the tasks increases for larger buoyancy force values, but U-Net based PDE surrogates show better interpolation and extrapolation abilities. In Figure 7, we display example scalar fields obtained for the parameter conditioning experiments of the Navier-Stokes equations using the best performing U-Net$_{\mathrm{mod}}$ model. Predicted and ground truth fields are shown for different values of the absolute buoyancy force and different time-scale values.

## 5 Conclusion

We presented a comprehensive comparison between various (Dilated) ResNet, FNO, and U-Net based approaches on fluid mechanic problems, paving a basis towards strong baselines for the development of neural PDE surrogates. For U-Nets, we transferred recent architectural improvements from computer vision, most notably from object segmentation and generative modeling. We found that the original U-Net architecture of Ronneberger et al. (2015) already functions as a powerful neural PDE surrogate, and e.g. outperforms FNOs on the presented tasks, which challenges prevailing claims in the literature, e.g. Li et al. (2020a). Combined with recent architectural improvements, we achieved significant performance improvements of U-Net based architectures. Secondly, we reported the first side by side comparison of neural PDE surrogates on vorticity-stream vs. velocity function formulation of fluid dynamics problems. Overall, velocity function formulations seem to be more challenging to model. Next, we used our insights on design considerations, and introduced U-FNets, i.e., modern U-Nets that are augmented with FNO downsampling layers. Those architectures further improve performance without major degradation of computational cost. As such, U-FNets are an appealing and well-motivated alternative in the vast design space of neural PDE surrogates containing the best of both worlds. Finally, we tested parameter conditioning techniques to train generalizable neural PDE surrogates. In doing so, we were able to show promising U-Net and FNO based architectures which generalize to different PDE parameters as well as different time-scales within a single surrogate model. However, similar to the findings of Lu et al. (2022) FNO layers seem to have negative effects when generalizing to different time-scales and PDE parameters. Finally, we hope that our codebase can be a starting point for further investigations on neural PDE surrogates.

**Limitations & Future Work.** This work focuses on the "image-to-image" modeling aspect of PDE surrogate modeling, more precisely on the understanding of complex multi-scale spatio-temporal phenomena. That said, in this work we did not elaborate on important aspects of neural PDE surrogates such as stability over long rollouts, preservation of invariants, or generalization over sampling regularities, over domain topologies and geometries, and over boundary conditions. We see many of these aspects as future work. Moreover, in this work we focused on modeling Navier-Stokes equations directly, rather then in the Reynolds-averaged Navier–Stokes (RANS) form (Tennekes et al., 1972), which is very common when describing turbulent flows. Finally, this work focuses on comprehensive comparison on various convolution based methods, whereas future work could extend towards Vision Transformers (Dosovitskiy et al., 2020), comparing their abilities to model and generalize across spatio-temporal information, especially with respect to data availability.

## Acknowledgements

We thank Alok Singh, Ratnesh Madaan, and Shuhang Chen for their comments on early versions of this paper. We are also thankful to Zongyi Li for suggesting extra settings for better comparisons of our FNO baselines.

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

# Contents

# A    Related work

Neural PDE modeling is appearing in many flavors. Various works are numerical-neural hybrid approaches where the computation graph of the solver is preserved and heuristically-chosen parameters are left for the neural network to predict (Bar-Sinai et al., 2019; Kochkov et al., 2021; Greenfeld et al., 2019; Hsieh et al., 2019; Praditia et al., 2021; Um et al., 2020; Garcia Satorras et al., 2019). The works of Sanchez-Gonzalez et al. (2020); Pfaff et al. (2020); Mayr et al. (2021) are of similar flavor where neural network predictions are input to the time-update of node positions in graphs and meshes. Fully neural network based approaches can be roughly split into two parts. First, methods that focus on the approximation of the solution function of the underlying PDE (Sirignano & Spiliopoulos, 2018; Han et al., 2018; Raissi et al., 2019; Jin et al., 2021; Raissi et al., 2020; Zubov et al., 2021). And second, methods that focus on the surrogate learning of solution operators. CNN-base models were among the first PDE surrogates (Guo et al., 2016; Bhatnagar et al., 2019; Zhu & Zabaras, 2018). Operator learning models were popularized via Fourier Neural Operators (Li et al., 2020a) and FNO-based applications and refinements (Li et al., 2021b; Rahman et al., 2022b; Rao et al., 2021; Guibas et al., 2021; Li et al., 2021a; Rahman et al., 2022a; Pathak et al., 2022; Wen et al., 2022; Liu et al., 2022; Yang et al., 2021; Guan et al., 2021; Hwang et al., 2022; Chen et al., 2021; Li et al., 2022b), as well as via DeepONets (Lu et al., 2019; 2021; 2022). Other directions include the modeling of PDE solution operators via latent space models, transformers, and graph neural networks (GNNs) (Wu et al., 2022; Li et al., 2022a; Brandstetter et al., 2022c; Lötzsch et al., 2022; Lienen & Günnemann, 2022). The ever persisting chicken-egg problem (Brandstetter et al., 2022b; Shi et al., 2022) of how to obtain high quality ground truth training data for neural PDE surrogates is approached either via clever data augmentation (Brandstetter et al., 2022b), via equivariant neural solvers (Wang et al., 2020b), or via "data-free" learning paradigms (Geneva & Zabaras, 2020; Wandel et al., 2020; 2022; Shi et al., 2022). Pratical applications of neural PDE surrogates can especially be found in weather forecasting (Pathak et al., 2022; Guibas et al., 2021; Keisler, 2022; Rasp & Thuerey, 2021; Weyn et al., 2020; 2021; Arcomano et al., 2020; Sønderby et al., 2020; Frerix et al., 2021; Maulik et al., 2022; Lam et al., 2022; Bi et al., 2022; Nguyen et al., 2023), and modeling of fluid dynamics (Ma et al., 2021; Stachenfeld et al., 2021; Wang et al., 2020a; Brandstetter et al., 2022a; Ruhe et al., 2023).

# B    Experiments

This appendix supports Section 4 of the main paper.

## B.1    Experimental details

**Loss functions and metrics.**   We report the summed MSE (SMSE) loss defined as:

$$\mathcal{L}_{\text{SMSE}} = \frac{1}{N_y} \sum_{y \in \mathbb{Z}^2} \sum_{j=1}^{N_t} \sum_{i=1}^{N_{\text{fields}}} \| \boldsymbol{u}_i(y, t_j) - \hat{\boldsymbol{u}}_i(y, t_j) \|_2^2 \ , \tag{8}$$

where $\boldsymbol{u}$ is the target, $\hat{\boldsymbol{u}}$ the model output, $N_{\text{fields}}$ comprises scalar fields as well as individual vector field components, and $N_y$ is the total number of spatial points. Equation 8 is used for training with $N_t = 1$, and further allows us to define our two main metrics:

- *One-step* loss where $N_t = 1$ and $N_{\text{fields}}$ comprises all scalar and vector components.

- *Rollout* loss where $N_t = 5$ and $N_{\text{fields}}$ comprises all scalar and vector components.

Alternatively, we train with the summed relative MSE (RMSE) loss as introduced in Li et al. (2020a):

$$\mathcal{L}_{\text{SMSE}} = \frac{1}{N_y} \sum_{y \in \mathbb{Z}^2} \sum_{j=1}^{N_t} \sum_{i=1}^{N_{\text{fields}}} \frac{\| \boldsymbol{u}_i(y, t_j) - \hat{\boldsymbol{u}}_i(y, t_j) \|_2^2}{\| \hat{\boldsymbol{u}}_i(y, t_j) \|_2^2} \ . \tag{9}$$

**Training and model selection.** We optimized models using the AdamW optimizer (Kingma & Ba, 2014; Loshchilov & Hutter, 2019) with the best learning rates of $[10^{-4}, 2 \cdot 10^{-4}]$ and weight decay of $10^{-5}$ for 50 epochs and minimized the summed mean squared error (SMSE) which is outlined in Equation 8. We used cosine annealing as learning rate scheduler (Loshchilov & Hutter, 2016) with a linear warmup. For baseline ResNet models, we optimized number of layers, number of channels, and normalization procedures. For the reported results we used group normalization (Wu & He, 2018) with 1 group which is equivalent to Layer norm (Ba et al., 2016) (except for final normalization layer in U-Nets where we use 8 groups). We further tested different activation functions. For baseline FNO models, we optimized number of layers, number of channels, and number of Fourier modes. Larger numbers of layers or channels did not improve the performances for both ResNet and FNO models. For U-Net like architectures, especially for U-Net$_{\text{att}}$, we specifically needed to optimize the maximum learning rate to be lower ($10^{-4}$). We further optimized for different number of hidden layers, and initialization and normalization schemes. For the reported results, we used pre-activations (He et al., 2016b) and layer normalization (Ba et al., 2016). We used an effective batch size of 32 for training.

**Computational resources.** All experiments used $4 \times 16$ GB NVIDIA V100 machines for training. Average training times varied between 2 h and 140 h, depending on task and number of trajectories. Parameter conditioning runs were the most expensive ones.

**Runtime comparison.** We warmup the benchmark for 10 iterations and report average runtimes over 100 runs on a single 16 GB NVIDIA V100 machine with input batch size of 8.

## B.2 Additional model details

### B.2.1 ResNet

We use two embedding and two output layers with kernel sizes of $1 \times 1$.

### B.2.2 Dilated ResNet

The implemented Dilated ResNet models consist of 4 residual blocks where each block individually consists of 7 dilated CNN layers with dilation rates of $[1, 2, 4, 8, 4, 2, 1]$. We implement Dilated ResNets with and without group normalization layers applied to each layer in the respective dilation blocks.

### B.2.3 FNO

We use FNOs consisting of $\{4, 8\}$ FNO layers, where $\{8, 16, 32\}$ modes are multiplied in the Fourier space, and $\{64, 128\}$ channels are used. We use two embedding and two output layers with kernel sizes of $1 \times 1$ as suggested in Li et al. (2020a). The number of non-zero Fourier modes, the number of FNO layers and the number of channels are hyperparameter, where we report results for different values in each of the experiment. We use GeLU activation functions, and no normalization scheme. Normalization schemes and residual connections did not improve performance, as already reported in Brandstetter et al. (2022a).

### B.2.4 U-Net

We use one embedding and one output layers with kernel sizes of $3 \times 3$. To allow a fair comparison to FNO (and ResNet) architectures, we ablated architectures also for kernel sizes of $1 \times 1$ for embedding and output layers.

**U-Net$_{2015}$.** We use channel multipliers of $(2, 2, 2, 2)$. The network consists overall of 4 downsampling, one bottleneck and 4 upsampling layers. We further use batch normalization (Ioffe & Szegedy, 2015), and no bias weights. The implementation is based on the `PDEbench` repository of Takamoto et al. (2022). Compared to the implementation of Takamoto et al. (2022), we use GeLU activations instead of tanh activations since we observe significant better performances. For the sake of completeness, we also report tanh results terming the models U-Net$_{2015\text{-}tanh}$.

**U-Net$_{\text{base}}$.** We use channel multipliers of $(2, 2, 2, 2)$. We replaced batch normalization (Ioffe & Szegedy, 2015) with group normalization (Wu & He, 2018) with number of groups equal 1 to be consistent with other architectures. Additionally, compared to the U-Net$_{2015}$ version, we use bias weights but no bottleneck layer.

**U-Net$_{\text{mod}}$.** We use channel multipliers of $(1, 2, 2, 4)$, and residual connections in each down- and upsampling block. We use pre-normalization and pre-activations (He et al., 2016b). Additionally, we zero-initialize the second `Conv` layer in each residual block.

**U-Net$_{\text{mod,attn}}$.** Adding attention to all down- and upsampling blocks made training unstable, and would have required an extensive hyperparameter search. We therefore only use attention in the middle blocks after downsampling. We further only use a single attention head along with a residual connection bypassing attention.

### B.2.5 Parameter Conditioning

**Embedding.** We use sinusoidal embedding as proposed in Vaswani et al. (2017) for positional encoding of scalar values, such as prediction time window and force strength:

$$\text{Emb}(x, d) = \left[\cos \frac{x}{10000^{2x/d_i}}, \sin \frac{x}{10000^{2x/d_i}}\right] \text{ for } 0 \leq d_i < d ,\qquad(10)$$

where $x$ is the embedded quantity and $d$ is the output embedding dimension.

**Projection.** We use a two-layer feed-forward network to project each of the embeddings to higher dimension ($4\times$ hidden channels), and add them together before passing them to each block via another linear layer.

**Conditioning.** We explore two different mechanisms for conditioning, originally proposed in the image modeling literature. Simple "Addition" as proposed by Ho et al. (2020) which can easily be extended to FNO layers and "AdaGN" as proposed by Nichol & Dhariwal (2021) which requires normalization layers to be applicable and was therefore restricted to U-Net based architectures in our experiments.

- **Addition:** A single `Linear` layer is used to scale the dimensions appropriately to match the dimensions of the conditioned block. Conditions are added to the first `Conv` layer's output, followed by normalization and a second `Conv` layer. This conditioning is applied to all blocks in the network.

- **AdaGN.** Instead of directly adding the `Linear` projection of the embedding to the respective blocks, the projection $y$ is split into $[y_s, y_b]$ to scale and shift the normalized output $h$ before passing to the second `Conv` layer in each block, similar to as is done in FiLM (Perez et al., 2018):

$$h' = y_s \odot \text{GroupNorm}(h) + y_b ,\qquad(11)$$

  where $\odot$ denotes the pointwise product. The conditioning is applied to all blocks. Since FNO architectures were implemented without group norm, AdaGN was not applicable to those.

In Appendix B.5, we ablate "Addition" and "AdaGN" embeddings for U-Net architectures.

### B.2.6 Spatial-spectral parameter conditioning for Fourier layers

Since FNO like architectures are usually implemented without normalization schemes, only "Addition" is applicable as conditioning strategy. Adding the conditioning at the end of each FNO layer, i.e. in the spatial domain, omits that the conditioning information is accessible in the Fourier domain too. This was somewhat unsatisfying, so we explored a straightforward mechanism to apply conditioning to the Fourier branch as well. We implement an alternative `FreqLinear` layer to project the embeddings into Fourier space too. Each Fourier mode is first multiplied with the embedding, then weights are mode-wise multiplied and the inverse Fourier transform is performed. Adding conditioning both in the Fourier and the spatial domain seems to work best. We term this alternative embedding "Spatial-Spectral" embedding. In Appendix B.5, we ablate "Addition" and "Spatial-Spectral" embeddings for FNO like architectures including UFNets with FNO blocks in the downsampling path.

### B.3 Shallow water equations.

The shallow water equations are solved on a regular grid with periodic boundary conditions as described in Section 4 of the main paper. The inputs to the shallow water experiments are respective fields at the previous 2 timesteps. Pressure and vorticity fields are normalized for training. Example rollout trajectories are displayed in Figure 8 for the velocity function formulation and in Figure 9 for the vorticity stream function formulation. We outline further details on the results on the shallow water experiments in Figures 10,11, and Tables 2,3. Additionally, we show results for 1-day predictions in Figure 12 and Table 7. We further ablate different encoding/decoding choices for U-Net like architectures in Figures 13,14. Finally, we compare different the specs of different FNO, UNO, and U-FNet architectures in Table 4.

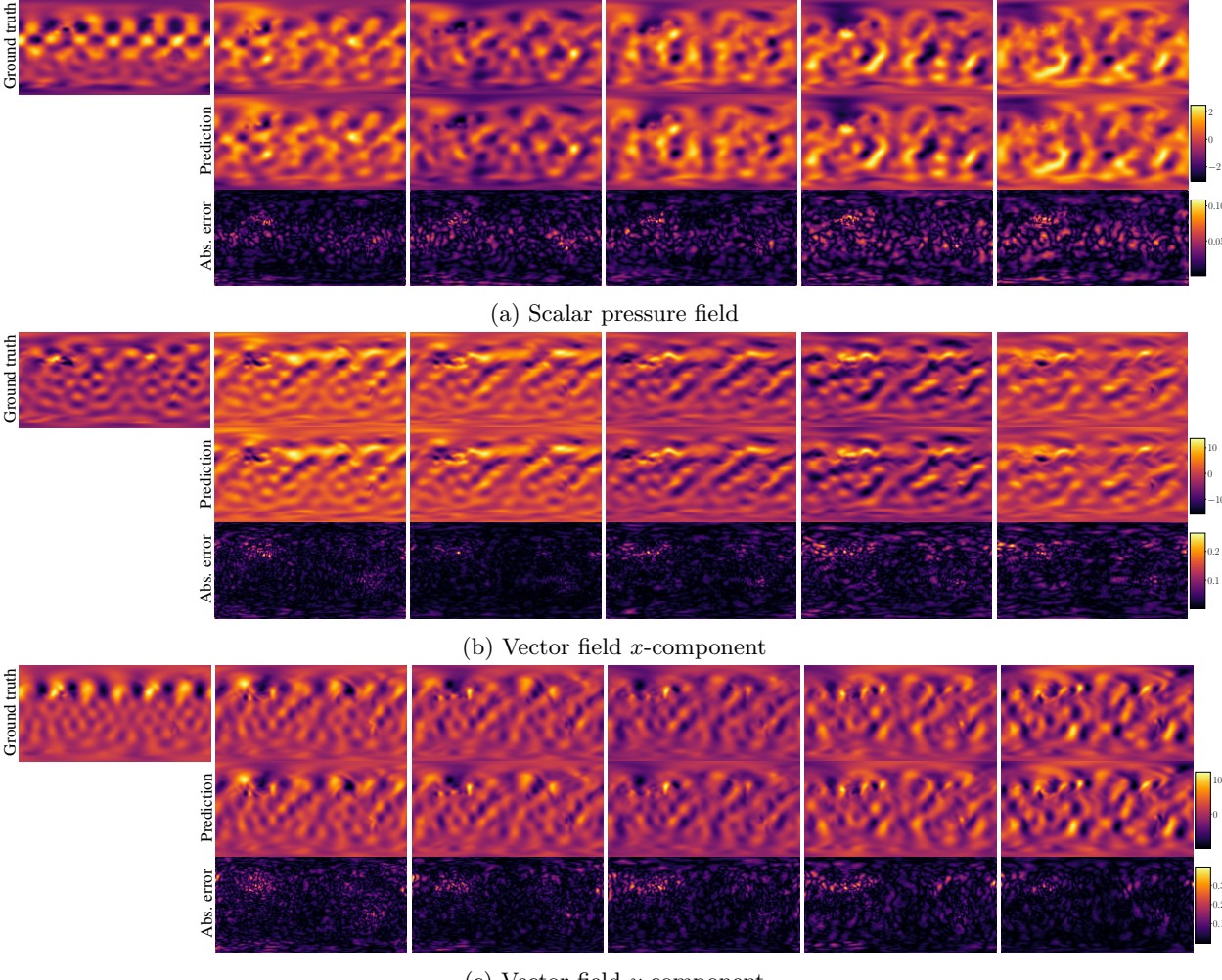

(a) Scalar pressure field

(b) Vector field $x$-component

(c) Vector field $y$-component

Figure 8: Shallow water 2-day predictions, velocity function form. Example rollouts of the scalar pressure and the vector wind field of the shallow water experiments are shown, obtained by a U-F1Net$_{\text{modes16}}$ PDE surrogate model (top), and compared to the ground truth (bottom). Predictions are obtained for a time window $\Delta t = 48\,\text{h}$. The respective model input fields comprise two timesteps, we only show the first of those (left-most ground truth column).

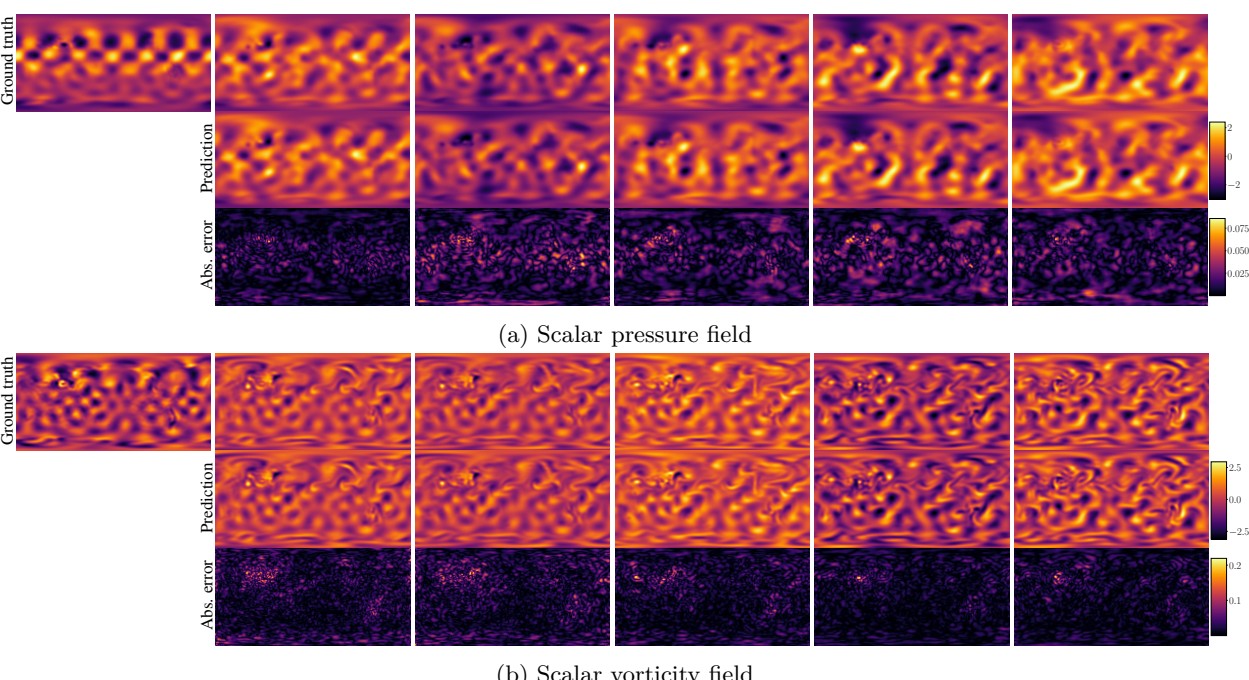

Figure 9: Shallow water 2-day predictions, vorticity stream function form. Example rollouts of the scalar pressure and the scalar vorticity field of the shallow water experiments are shown, obtained by a U-F2Net$_{\mathrm{modes16,16}}$ PDE surrogate model (top), and compared to the ground truth (bottom). Predictions are obtained for a time window $\Delta t = 48\,\mathrm{h}$. The respective model input fields comprise two timesteps, we only show the first of those (left-most ground truth column).

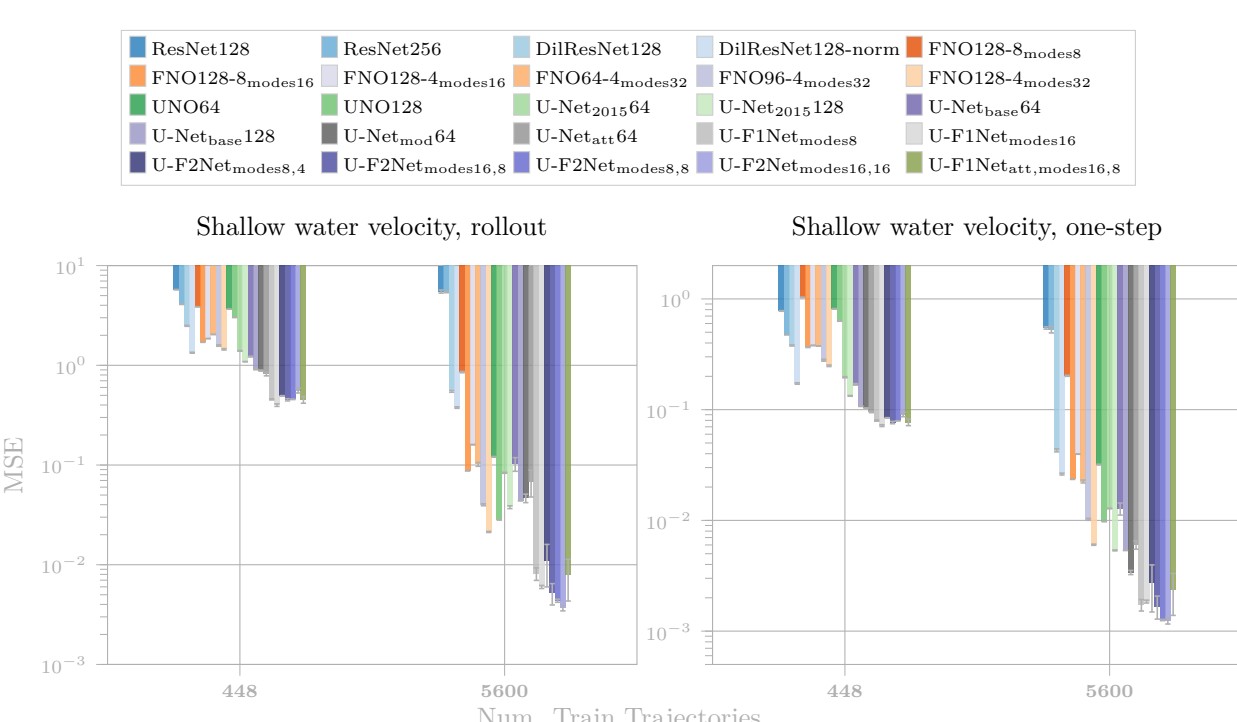

Figure 10: Shallow water 2-day predictions. Rollout and one-step errors of various architectures on the shallow water equations are reported. Results are obtained for predictions of 2-day time windows for the velocity function formulation and are averaged over three different random seeds. Note the logarithmic scale of the $y$-axes.

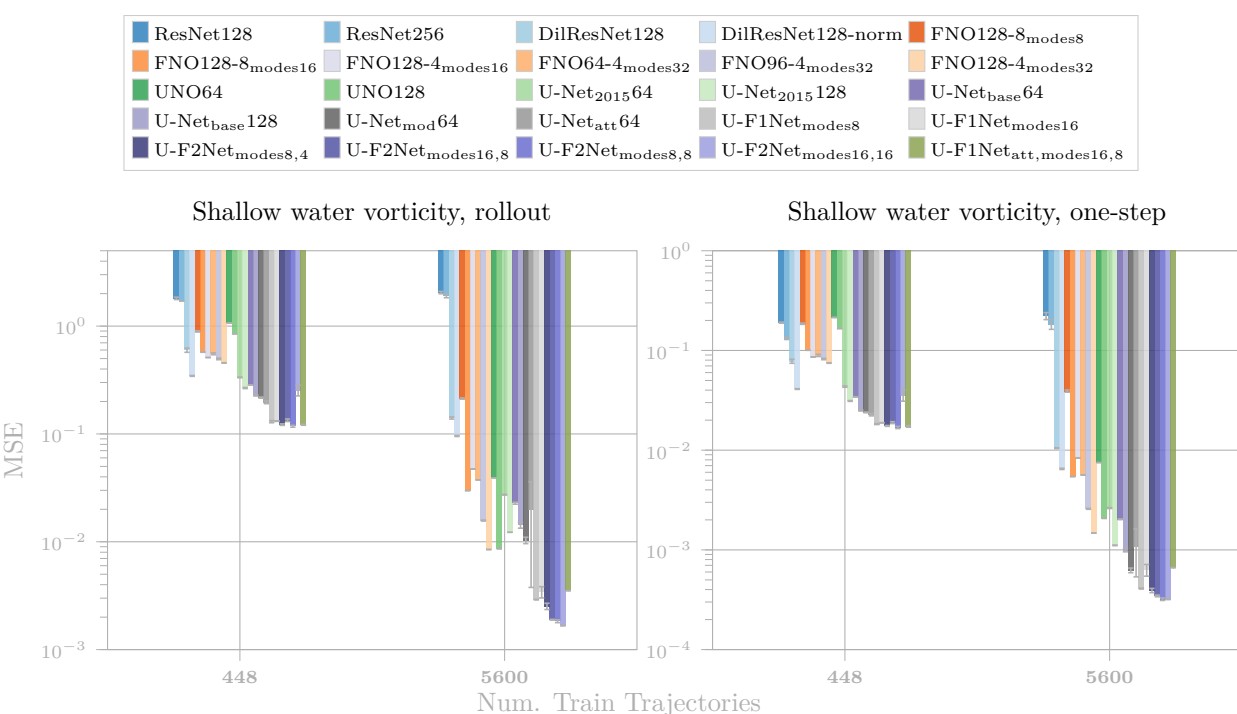

Figure 11: Shallow water 2-day predictions. Rollout and one-step errors of various architectures on the shallow water equations. Results are obtained for predictions of 2-day time windows for the vorticity stream function formulation and are averaged over three different random seeds. Note the logarithmic scale of the $y$-axes.

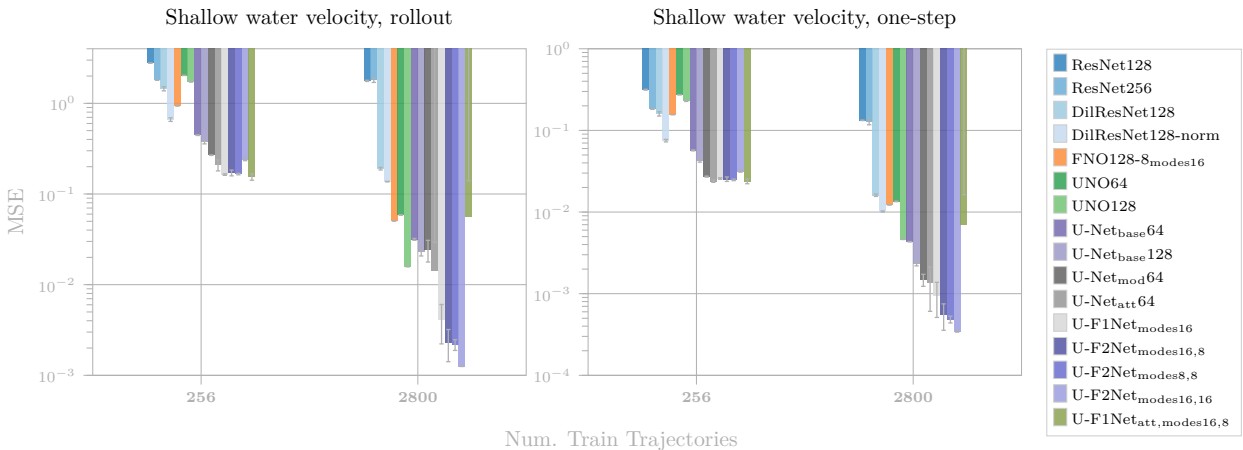

Figure 12: Shallow water 1-day predictions. Rollout and one-step errors of various architectures on the shallow water equations are reported. Results are obtained for predictions of 1-day time windows for the velocity function formulation and are averaged over three different random seeds. Note the logarithmic scale of the $y$-axes.

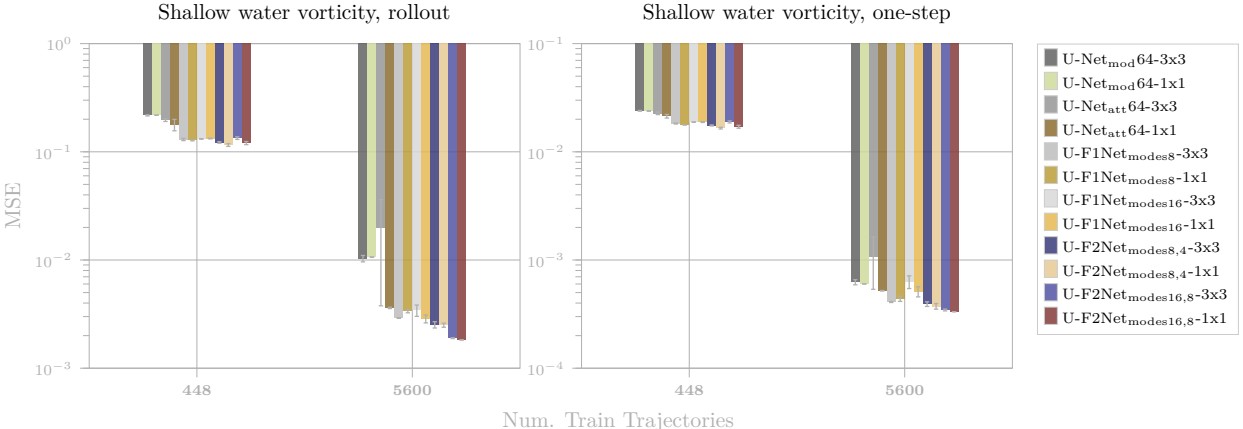

Figure 13: Shallow water 2-day predictions. Ablation results of different encoding and decoding choices for various U-Net architectures are reported. $1 \times 1$ and $3 \times 3$ kernels are compared for both encoding and decoding. Rollout and one-step errors are obtained on the shallow water equations for 2-day predictions for the vorticity stream function formulation and are averaged over three different random seeds.

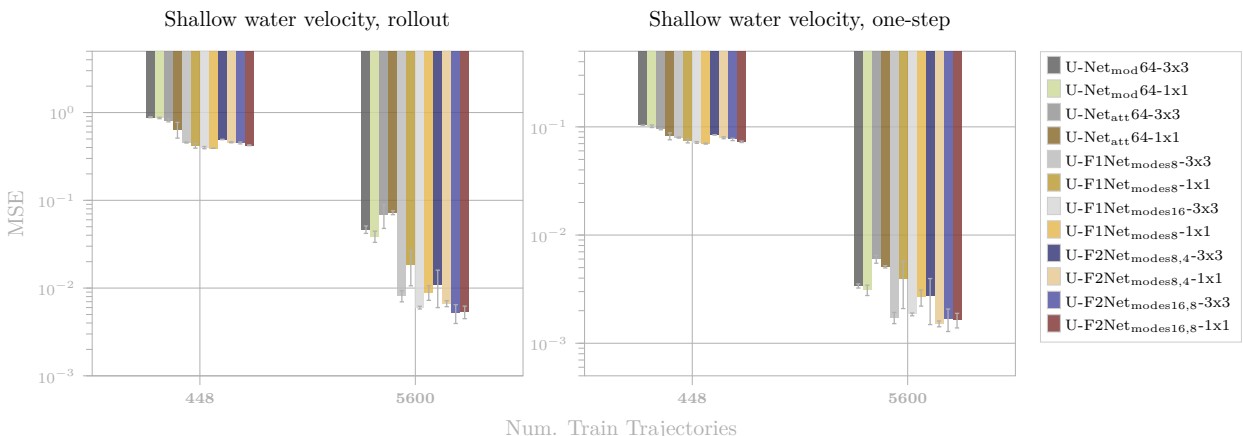

Figure 14: Shallow water 2-day predictions. Ablation results of different encoding and decoding choices for various U-Net architectures are reported. $1 \times 1$ and $3 \times 3$ kernels are compared for both encoding and decoding. Rollout and one-step errors are obtained on the shallow water equations for 2-day predictions for the velocity function formulation and are averaged over three different random seeds.

Table 2: Shallow water 2-day predictions, velocity function formulation. Rollout and one-step errors of various architectures on the shallow water equations are reported. Summed mean-squared errors (SMSE) are obtained for 2-day predictions for the velocity function formulation and are averaged over three different random seeds. If results are displayed without standard deviation, the obtained standard deviation is lower than the four digit precision minimum. The best model of each model class is highlighted.

| | | SMSE | |
|---|---|---|---|
| METHOD | Trajs. | onestep | rollout |
| ResNet128 | 448 | $0.7787 \pm 0.0049$ | $5.7408 \pm 0.0223$ |
| ResNet128 | 5600 | $0.5465 \pm 0.0130$ | $5.4946 \pm 0.2009$ |
| ResNet256 | 448 | $0.4751 \pm 0.0005$ | $4.0954 \pm 0.0181$ |
| ResNet256 | 5600 | $0.5294 \pm 0.0364$ | $5.4155 \pm 0.0657$ |
| DilResNet128 | 448 | $0.3800 \pm 0.0042$ | $2.4860 \pm 0.0260$ |
| DilResNet128 | 5600 | $0.0429 \pm 0.0014$ | $0.5476 \pm 0.0109$ |
| DilResNet128-norm | 448 | $0.1723 \pm 0.0026$ | $1.3377 \pm 0.0144$ |
| DilResNet128-norm | 5600 | $0.0262 \pm 0.0006$ | $0.3770 \pm 0.0081$ |
| FNO128-8$_{modes8}$ | 448 | $1.0322 \pm 0.0055$ | $3.8635 \pm 0.0090$ |
| FNO128-8$_{modes8}$ | 5600 | $0.2023 \pm 0.0023$ | $0.8549 \pm 0.0124$ |
| FNO128-8$_{modes16}$ | 448 | $0.3681 \pm 0.0042$ | $1.7088 \pm 0.0096$ |
| FNO128-8$_{modes16}$ | 5600 | $0.0236 \pm 0.0001$ | $0.0878 \pm 0.0007$ |
| FNO128-4$_{modes16}$ | 448 | $0.3802 \pm 0.0021$ | $1.8542 \pm 0.0056$ |
| FNO128-4$_{modes16}$ | 5600 | $0.0397 \pm 0.0002$ | $0.1601 \pm 0.0010$ |
| FNO64-4$_{modes32}$ | 448 | $0.3750 \pm 0.0012$ | $2.0393 \pm 0.0050$ |
| FNO64-4$_{modes32}$ | 5600 | $0.0225 \pm 0.0007$ | $0.1015 \pm 0.0044$ |
| FNO96-4$_{modes32}$ | 448 | $0.2794 \pm 0.0053$ | $1.5788 \pm 0.0253$ |
| FNO96-4$_{modes32}$ | 5600 | $0.0102 \pm 0.0002$ | $0.0399 \pm 0.0008$ |
| FNO128-4$_{modes32}$ | 448 | $0.2492 \pm 0.0040$ | $1.4460 \pm 0.0226$ |
| FNO128-4$_{modes32}$ | 5600 | $0.0060 \pm 0.0001$ | $0.0213 \pm 0.0003$ |
| UNO64 | 448 | $0.8134 \pm 0.0048$ | $3.6621 \pm 0.0103$ |
| UNO64 | 5600 | $0.0319 \pm 0.0003$ | $0.1208 \pm 0.0010$ |
| UNO128 | 448 | $0.6328 \pm 0.0041$ | $3.0240 \pm 0.0064$ |
| UNO128 | 5600 | $0.0098 \pm 0.0001$ | $0.0282 \pm 0.0002$ |
| U-Net$_{base}$64 | 448 | $0.1693 \pm 0.0026$ | $1.2224 \pm 0.0108$ |
| U-Net$_{base}$64 | 5600 | $0.0128 \pm 0.0016$ | $0.1026 \pm 0.0161$ |
| U-Net$_{base}$128 | 448 | $0.1076 \pm 0.0008$ | $0.9096 \pm 0.0067$ |
| U-Net$_{base}$128 | 5600 | $0.0054$ | $0.0439 \pm 0.0001$ |
| U-Net$_{mod}$64 | 448 | $0.1034 \pm 0.0001$ | $0.8847 \pm 0.0109$ |
| U-Net$_{mod}$64 | 5600 | $0.0034 \pm 0.0001$ | $0.0465 \pm 0.0045$ |
| U-Net$_{mod}$64-1x1 | 448 | $0.1013 \pm 0.0028$ | $0.8681 \pm 0.0161$ |
| U-Net$_{mod}$64-1x1 | 5600 | $0.0031 \pm 0.0003$ | $0.0389 \pm 0.0057$ |
| U-Net$_{att}$64 | 448 | $0.0954 \pm 0.0014$ | $0.8158 \pm 0.0318$ |
| U-Net$_{att}$64 | 5600 | $0.0060 \pm 0.0005$ | $0.0684 \pm 0.0206$ |
| U-Net$_{att}$64-1x1 | 448 | $0.0819 \pm 0.0060$ | $0.6419 \pm 0.1307$ |
| U-Net$_{att}$64-1x1 | 5600 | $0.0051 \pm 0.0001$ | $0.0724 \pm 0.0037$ |
| U-F1Net$_{modes8}$ | 448 | $0.0797 \pm 0.0010$ | $0.4553 \pm 0.0057$ |
| U-F1Net$_{modes8}$ | 5600 | $0.0017 \pm 0.0002$ | $0.0081 \pm 0.0012$ |
| U-F1Net$_{modes16}$ | 448 | $0.0717 \pm 0.0013$ | $0.3988 \pm 0.0115$ |
| U-F1Net$_{modes16}$ | 5600 | $0.0018 \pm 0.0001$ | $0.0060 \pm 0.0002$ |
| U-F1Net$_{modes8}$-1x1 | 448 | $0.0743 \pm 0.0032$ | $0.4185 \pm 0.0259$ |
| U-F1Net$_{modes8}$-1x1 | 5600 | $0.0039 \pm 0.0018$ | $0.0185 \pm 0.0079$ |
| U-F1Net$_{modes16}$-1x1 | 448 | $0.0699 \pm 0.0007$ | $0.3923 \pm 0.0001$ |
| U-F1Net$_{modes16}$-1x1 | 5600 | $0.0027 \pm 0.0004$ | $0.0089 \pm 0.0017$ |
| U-F2Net$_{modes8,4}$ | 448 | $0.0843 \pm 0.0004$ | $0.4934 \pm 0.0031$ |
| U-F2Net$_{modes8,4}$ | 5600 | $0.0027 \pm 0.0012$ | $0.0110 \pm 0.0050$ |
| U-F2Net$_{modes16,8}$ | 448 | $0.0765 \pm 0.0019$ | $0.4508 \pm 0.0116$ |
| U-F2Net$_{modes16,8}$ | 5600 | $0.0017 \pm 0.0004$ | $0.0052 \pm 0.0013$ |
| U-F2Net$_{modes8,8}$ | 448 | $0.0793 \pm 0.0001$ | $0.4553 \pm 0.0021$ |
| U-F2Net$_{modes8,8}$ | 5600 | $0.0013$ | $0.0044 \pm 0.0002$ |
| U-F2Net$_{modes16,16}$ | 448 | $0.0906 \pm 0.0042$ | $0.5596 \pm 0.0317$ |
| U-F2Net$_{modes16,16}$ | 5600 | $0.0012 \pm 0.0001$ | $0.0037 \pm 0.0002$ |
| U-F2Net$_{modes8,4}$-1x1 | 448 | $0.0793 \pm 0.0016$ | $0.4587 \pm 0.0064$ |
| U-F2Net$_{modes8,4}$-1x1 | 5600 | $0.0015 \pm 0.0001$ | $0.0067 \pm 0.0005$ |
| U-F2Net$_{modes16,8}$-1x1 | 448 | $0.0728 \pm 0.0012$ | $0.4238 \pm 0.0049$ |
| U-F2Net$_{modes16,8}$-1x1 | 5600 | $0.0016 \pm 0.0003$ | $0.0054 \pm 0.0009$ |
| U-F2Net$_{att,modes16,8}$ | 448 | $0.0769 \pm 0.0052$ | $0.4517 \pm 0.0348$ |
| U-F2Net$_{att,modes16,8}$ | 5600 | $0.0024 \pm 0.0010$ | $0.0078 \pm 0.0035$ |
| U-F3Net$_{modes8,4,2}$ | 448 | $0.0920 \pm 0.0011$ | $0.5318 \pm 0.0083$ |
| U-F3Net$_{modes8,4,2}$ | 5600 | $0.0018$ | $0.0062 \pm 0.0001$ |
| U-F3Net$_{modes16,8,4}$ | 448 | $0.0812$ | $0.4823 \pm 0.0021$ |
| U-F3Net$_{modes16,8,4}$ | 5600 | $0.0016$ | $0.0048$ |

Table 3: Shallow water 2-day predictions, vorticity stream function formulation. Rollout and one-step errors of various architectures on the shallow water equations are reported. Summed mean-squared errors (SMSE) are obtained for 2-day predictions for the vorticity stream function formulation and are averaged over three different random seeds. If results are displayed without standard deviation, the obtained standard deviation is lower than the five digit precision minimum. The best model of each model class is highlighted.

| Method | Trajs. | SMSE onestep | SMSE rollout |
|---|---|---|---|
| ResNet128 | 448 | 0.18993 ± 0.00232 | 1.80930 ± 0.04636 |
| ResNet128 | 5600 | 0.22182 ± 0.01793 | 2.04572 ± 0.05028 |
| ResNet256 | 448 | 0.13208 ± 0.00325 | 1.71041 ± 0.00142 |
| ResNet256 | 5600 | 0.18380 ± 0.02108 | 1.90615 ± 0.07268 |
| DilResNet128 | 448 | 0.07797 ± 0.00360 | 0.59831 ± 0.02661 |
| DilResNet128 | 5600 | 0.01047 ± 0.00011 | 0.14051 ± 0.00320 |
| DilResNet128-norm | 448 | 0.04105 ± 0.00035 | 0.34523 ± 0.00303 |
| DilResNet128-norm | 5600 | 0.00649 ± 0.00009 | 0.09571 ± 0.00110 |
| FNO128-8$_{modes8}$ | 448 | 0.18553 ± 0.00278 | 0.89244 ± 0.01109 |
| FNO128-8$_{modes8}$ | 5600 | 0.03917 ± 0.00101 | 0.21304 ± 0.00281 |
| FNO128-8$_{modes16}$ | 448 | 0.10054 ± 0.00026 | 0.57404 ± 0.00240 |
| FNO128-8$_{modes16}$ | 5600 | 0.00544 ± 0.00001 | 0.02981 ± 0.00005 |
| FNO128-4$_{modes16}$ | 448 | 0.08601 ± 0.00028 | 0.51127 ± 0.00041 |
| FNO128-4$_{modes16}$ | 5600 | 0.00836 ± 0.00003 | 0.04723 ± 0.00005 |
| FNO64-4$_{modes32}$ | 448 | 0.08923 ± 0.00233 | 0.55383 ± 0.01140 |
| FNO64-4$_{modes32}$ | 5600 | 0.00564 ± 0.00002 | 0.03744 |
| FNO96-4$_{modes32}$ | 448 | 0.08208 ± 0.00132 | 0.49644 ± 0.00851 |
| FNO96-4$_{modes32}$ | 5600 | 0.00258 ± 0.00003 | 0.01573 ± 0.00016 |
| FNO128-4$_{modes32}$ | 448 | 0.07483 ± 0.00049 | 0.45585 ± 0.00224 |
| FNO128-4$_{modes32}$ | 5600 | 0.00147 | 0.00847 ± 0.00003 |
| UNO64 | 448 | 0.21437 | 1.06814 |
| UNO64 | 5600 | 0.00751 ± 0.00001 | 0.03918 ± 0.00002 |
| UNO128 | 448 | 0.16632 ± 0.00140 | 0.84857 ± 0.00190 |
| UNO128 | 5600 | 0.00207 | 0.00858 |
| U-Net$_{base}$64 | 448 | 0.03422 ± 0.00014 | 0.28432 ± 0.00016 |
| U-Net$_{base}$64 | 5600 | 0.00203 ± 0.00002 | 0.02277 ± 0.00047 |
| U-Net$_{base}$128 | 448 | 0.02505 ± 0.00035 | 0.22793 ± 0.00297 |
| U-Net$_{base}$128 | 5600 | 0.00099 ± 0.00004 | 0.01425 ± 0.00089 |
| U-Net$_{mod}$64 | 448 | 0.02393 ± 0.00020 | 0.21713 ± 0.00254 |
| U-Net$_{mod}$64 | 5600 | 0.00062 ± 0.00003 | 0.01031 ± 0.00070 |
| U-Net$_{mod}$64-1x1 | 448 | 0.02383 ± 0.00011 | 0.21811 ± 0.00045 |
| U-Net$_{mod}$64-1x1 | 5600 | 0.00060 | 0.01063 ± 0.00006 |
| U-Net$_{att}$64 | 448 | 0.02269 ± 0.00060 | 0.19643 ± 0.00566 |
| U-Net$_{att}$64 | 5600 | 0.00108 ± 0.00055 | 0.01994 ± 0.01617 |
| U-Net$_{att}$64-1x1 | 448 | 0.02133 ± 0.00083 | 0.17791 ± 0.02144 |
| U-Net$_{att}$64-1x1 | 5600 | 0.00052 | 0.00361 ± 0.00005 |
| U-F1Net$_{modes8}$ | 448 | 0.01827 ± 0.00020 | 0.12995 ± 0.00297 |
| U-F1Net$_{modes8}$ | 5600 | 0.00041 | 0.00291 ± 0.00002 |
| U-F1Net$_{modes16}$ | 448 | 0.01882 ± 0.00007 | 0.13156 ± 0.00074 |
| U-F1Net$_{modes16}$ | 5600 | 0.00063 ± 0.00008 | 0.00342 ± 0.00041 |
| U-F1Net$_{modes8}$-1x1 | 448 | 0.01774 ± 0.00004 | 0.12751 ± 0.00120 |
| U-F1Net$_{modes8}$-1x1 | 5600 | 0.00044 ± 0.00002 | 0.00336 ± 0.00011 |
| U-F1Net$_{modes16}$-1x1 | 448 | 0.01882 ± 0.00014 | 0.13287 ± 0.00127 |
| U-F1Net$_{modes16}$-1x1 | 5600 | 0.00051 ± 0.00005 | 0.00286 ± 0.00024 |
| U-F2Net$_{modes8,4}$ | 448 | 0.01751 ± 0.00019 | 0.12203 ± 0.00199 |
| U-F2Net$_{modes8,4}$ | 5600 | 0.00039 ± 0.00002 | 0.00252 ± 0.00017 |
| U-F2Net$_{modes8,4}$-1x1 | 448 | 0.01656 ± 0.00030 | 0.11567 ± 0.00336 |
| U-F2Net$_{modes8,4}$-1x1 | 5600 | 0.00037 ± 0.00002 | 0.00250 ± 0.00011 |
| U-F2Net$_{modes16,8}$ | 448 | 0.01890 ± 0.00042 | 0.13386 ± 0.00363 |
| U-F2Net$_{modes16,8}$ | 5600 | 0.00035 ± 0.00001 | 0.00189 ± 0.00002 |
| U-F2Net$_{modes16,8}$-1x1 | 448 | 0.01705 ± 0.00052 | 0.12052 ± 0.00404 |
| U-F2Net$_{modes16,8}$-1x1 | 5600 | 0.00033 | 0.00183 |
| U-F2Net$_{modes8,8}$ | 448 | 0.01689 ± 0.00043 | 0.11781 ± 0.00235 |
| U-F2Net$_{modes8,8}$ | 5600 | 0.00032 ± 0.00001 | 0.00183 ± 0.00005 |
| U-F2Net$_{modes16,16}$ | 448 | 0.03566 ± 0.00467 | 0.25400 ± 0.02869 |
| U-F2Net$_{modes16,16}$ | 5600 | 0.00032 | 0.00167 ± 0.00001 |
| U-F2Net$_{att,modes16,8}$ | 448 | 0.01731 ± 0.00031 | 0.12150 ± 0.00119 |
| U-F2Net$_{att,modes16,8}$ | 5600 | 0.00066 | 0.00352 |
| U-F3Net$_{modes8,4,2}$ | 448 | 0.02058 ± 0.00013 | 0.14738 ± 0.00131 |
| U-F3Net$_{modes8,4,2}$ | 5600 | 0.00040 | 0.00237 ± 0.00001 |
| U-F3Net$_{modes16,8,4}$ | 448 | 0.02203 ± 0.00026 | 0.15855 ± 0.00055 |
| U-F3Net$_{modes16,8,4}$ | 5600 | 0.00038 ± 0.00001 | 0.00208 ± 0.00005 |

Table 4: Comparison of parameter count, runtime, and memory requirement of various FNO, UNO, and U-FNet architectures. Subscript numbers indicate the used number of Fourier modes. For U-FNet experiments subscript numbers indicate the number of Fourier modes in the lowest, second-lowest, and third lowest block.

| Method | Channels | Res.Layers/Blocks | Params. | Runtime [s] | | Mem. [MB] | |
|---|---|---|---|---|---|---|---|
| | | | | Fwd. | Fwd.+bwd. | f32 size | Peak usage |
| FNO128-8$_{\text{modes8}}$ | 128 | 8 | 33.7 M | 0.057 | 0.162 | 134 | 2161 |
| FNO128-8$_{\text{modes16}}$ | 128 | 8 | 134 M | 0.059 | 0.171 | 537 | 2953 |
| FNO128-4$_{\text{modes16}}$ | 128 | 4 | 67.2 M | 0.031 | 0.089 | 268 | 1852 |
| FNO64-4$_{\text{modes32}}$ | 64 | 4 | 67.1 M | 0.016 | 0.050 | 268 | 1204 |
| FNO96-4$_{\text{modes32}}$ | 96 | 4 | 151 M | 0.026 | 0.080 | 604 | 2179 |
| FNO128-4$_{\text{modes32}}$ | 128 | 4 | 268 M | 0.036 | 0.118 | 1100 | 3420 |
| UNO64 | 64 | 7 | 110 M | 0.070 | 0.134 | 440 | 1925 |
| UNO128 | 128 | 7 | 440 M | 0.160 | 0.341 | 1800 | 5513 |
| U-F1Net$_{\text{modes8}}$ | 64 | 9 | 154 M | 0.083 | 0.205 | 617 | 3936 |
| U-F1Net$_{\text{modes16}}$ | 64 | 9 | 185 M | 0.084 | 0.208 | 743 | 4037 |
| U-F2Net$_{\text{modes8,4}}$ | 64 | 9 | 163 M | 0.085 | 0.213 | 652 | 3961 |
| U-F2Net$_{\text{modes8,8}}$ | 64 | 9 | 193 M | 0.085 | 0.216 | 772 | 4046 |
| U-F2Net$_{\text{modes16,8}}$ | 64 | 9 | 224 M | 0.086 | 0.219 | 897 | 4149 |
| U-F2Net$_{\text{modes16,16}}$ | 64 | 9 | 344 M | 0.090 | 0.232 | 1400 | 4496 |
| U-F3Net$_{\text{modes8,4,2}}$ | 64 | 9 | 198 M | 0.086 | 0.221 | 658 | 4332 |
| U-F3Net$_{\text{modes16,8,4}}$ | 64 | 9 | 259 M | 0.088 | 0.226 | 1000 | 4808 |

Table 5: Shallow water 2-day predictions, velocity function formulation. Rollout and one-step errors of various architectures on the shallow water equations are reported. L2 training objective of Li et al. (2020a) is used. Summed mean-squared errors (SMSE) are obtained for 2-day predictions for the vorticity stream function formulation and are averaged over three different random seeds. If results are displayed without standard deviation, the obtained standard deviation is lower than the five digit precision minimum. The best model of each model class is highlighted.

| METHOD | Trajs. | SMSE | |
| --- | --- | --- | --- |
| | | onestep | rollout |
| DilResNet128 | 448 | $0.312\,43 \pm 0.004\,90$ | $2.115\,97 \pm 0.036\,31$ |
| DilResNet128 | 5600 | $0.042\,30 \pm 0.000\,75$ | $0.542\,99 \pm 0.012\,92$ |
| DilResNet128-norm | 448 | $0.148\,49 \pm 0.002\,38$ | $1.197\,04 \pm 0.008\,19$ |
| DilResNet128-norm | 5600 | $0.026\,07 \pm 0.000\,73$ | $0.382\,68 \pm 0.007\,60$ |
| FNO128-4$_{\text{modes16}}$ | 448 | $0.339\,75 \pm 0.000\,37$ | $1.673\,92 \pm 0.003\,99$ |
| FNO128-4$_{\text{modes16}}$ | 5600 | $0.040\,21 \pm 0.000\,49$ | $0.163\,17 \pm 0.002\,58$ |
| FNO64-4$_{\text{modes32}}$ | 448 | $0.302\,18 \pm 0.002\,98$ | $1.678\,19 \pm 0.015\,23$ |
| FNO64-4$_{\text{modes32}}$ | 5600 | $0.023\,08 \pm 0.001\,02$ | $0.107\,56 \pm 0.006\,50$ |
| FNO96-4$_{\text{modes32}}$ | 448 | $0.234\,39 \pm 0.005\,60$ | $1.351\,64 \pm 0.025\,51$ |
| FNO96-4$_{\text{modes32}}$ | 5600 | $0.010\,16 \pm 0.000\,31$ | $0.040\,49 \pm 0.001\,50$ |
| U-Net$_{\text{base}}$64 | 448 | $0.147\,26 \pm 0.002\,30$ | $1.115\,39 \pm 0.005\,14$ |
| U-Net$_{\text{base}}$64 | 5600 | $0.009\,88 \pm 0.000\,11$ | $0.067\,11 \pm 0.001\,57$ |
| U-Net$_{\text{base}}$128 | 448 | $0.095\,07 \pm 0.000\,22$ | $0.844\,98 \pm 0.002\,95$ |
| U-Net$_{\text{base}}$128 | 5600 | $0.004\,35 \pm 0.000\,08$ | $0.032\,08 \pm 0.000\,44$ |
| U-Net$_{\text{2015}}$64 | 448 | $0.169\,45 \pm 0.001\,13$ | $1.264\,75 \pm 0.003\,96$ |
| U-Net$_{\text{2015}}$64 | 5600 | $0.012\,79 \pm 0.000\,09$ | $0.084\,20 \pm 0.001\,29$ |
| U-Net$_{\text{2015}}$128 | 448 | $0.114\,96 \pm 0.000\,23$ | $0.985\,53 \pm 0.001\,19$ |
| U-Net$_{\text{2015}}$128 | 5600 | $0.005\,41 \pm 0.000\,06$ | $0.039\,10 \pm 0.001\,39$ |
| U-Net$_{\text{2015-tanh}}$64 | 448 | $0.475\,99 \pm 0.002\,45$ | $2.760\,14 \pm 0.004\,88$ |
| U-Net$_{\text{2015-tanh}}$64 | 5600 | $0.029\,97 \pm 0.001\,37$ | $0.190\,94 \pm 0.006\,18$ |
| U-Net$_{\text{2015-tanh}}$128 | 448 | $0.333\,82 \pm 0.004\,30$ | $2.187\,05 \pm 0.025\,79$ |
| U-Net$_{\text{2015-tanh}}$128 | 5600 | $0.015\,05 \pm 0.000\,32$ | $0.097\,31 \pm 0.001\,29$ |
| U-Net$_{\text{mod}}$64 | 448 | $0.088\,51 \pm 0.001\,42$ | $0.808\,42 \pm 0.011\,27$ |
| U-Net$_{\text{mod}}$64 | 5600 | $0.002\,25 \pm 0.000\,07$ | $0.021\,95 \pm 0.001\,11$ |
| U-F2Net$_{\text{modes16,8}}$ | 448 | $0.061\,98 \pm 0.000\,23$ | $0.371\,33 \pm 0.001\,92$ |
| U-F2Net$_{\text{modes16,8}}$ | 5600 | $0.001\,23 \pm 0.000\,02$ | $0.003\,76 \pm 0.000\,04$ |

Table 6: Shallow water 2-day predictions, vorticity stream function formulation. Rollout and one-step errors of various architectures on the shallow water equations are reported. L2 training objective of Li et al. (2020a) is used. Summed mean-squared errors (SMSE) are obtained for 2-day predictions for the vorticity stream function formulation and are averaged over three different random seeds. If results are displayed without standard deviation, the obtained standard deviation is lower than the five digit precision minimum. The best model of each model class is highlighted.

| | | SMSE | |
|---|---|---|---|
| Method | Trajs. | onestep | rollout |
| DilResNet128 | 448 | $0.069\,14 \pm 0.004\,30$ | $0.541\,08 \pm 0.034\,62$ |
| DilResNet128 | 5600 | $0.010\,49 \pm 0.000\,19$ | $0.141\,19 \pm 0.005\,32$ |
| DilResNet128-norm | 448 | $0.036\,62 \pm 0.000\,21$ | $0.314\,18 \pm 0.002\,97$ |
| DilResNet128-norm | 5600 | $0.006\,59 \pm 0.000\,11$ | $0.095\,90 \pm 0.000\,95$ |
| FNO128-8$_{modes8}$ | 448 | $0.178\,17 \pm 0.004\,62$ | $0.864\,18 \pm 0.019\,15$ |
| FNO128-8$_{modes8}$ | 5600 | $0.039\,73 \pm 0.000\,99$ | $0.213\,97 \pm 0.002\,21$ |
| FNO128-8$_{modes16}$ | 448 | $0.093\,02 \pm 0.003\,25$ | $0.543\,09 \pm 0.016\,52$ |
| FNO128-8$_{modes16}$ | 5600 | $0.005\,10 \pm 0.000\,02$ | $0.027\,47 \pm 0.000\,04$ |
| FNO64-4$_{modes32}$ | 448 | $0.080\,56 \pm 0.001\,65$ | $0.505\,15 \pm 0.008\,60$ |
| FNO64-4$_{modes32}$ | 5600 | $0.005\,63 \pm 0.000\,01$ | $0.037\,19 \pm 0.000\,09$ |
| FNO96-4$_{modes32}$ | 448 | $0.074\,53 \pm 0.001\,33$ | $0.458\,38 \pm 0.007\,61$ |
| FNO96-4$_{modes32}$ | 5600 | $0.002\,48 \pm 0.000\,04$ | $0.014\,81 \pm 0.000\,23$ |
| FNO128-4$_{modes32}$ | 448 | $0.067\,41 \pm 0.000\,71$ | $0.418\,39 \pm 0.003\,05$ |
| FNO128-4$_{modes32}$ | 5600 | $0.001\,33 \pm 0.000\,01$ | $0.007\,42 \pm 0.000\,01$ |
| UNO64 | 448 | $0.192\,35 \pm 0.000\,10$ | $0.973\,45 \pm 0.001\,86$ |
| UNO64 | 5600 | $0.007\,29 \pm 0.000\,01$ | $0.037\,24 \pm 0.000\,10$ |
| U-Net$_{base}$64 | 448 | $0.030\,15 \pm 0.000\,12$ | $0.258\,04 \pm 0.001\,29$ |
| U-Net$_{base}$64 | 5600 | $0.001\,99 \pm 0.000\,02$ | $0.021\,40 \pm 0.000\,17$ |
| U-Net$_{base}$128 | 448 | $0.021\,85 \pm 0.000\,20$ | $0.206\,57 \pm 0.001\,45$ |
| U-Net$_{base}$128 | 5600 | $0.000\,84 \pm 0.000\,02$ | $0.010\,85 \pm 0.000\,49$ |
| U-Net$_{2015}$64 | 448 | $0.038\,41 \pm 0.000\,52$ | $0.307\,50 \pm 0.003\,27$ |
| U-Net$_{2015}$64 | 5600 | $0.002\,66 \pm 0.000\,02$ | $0.027\,44 \pm 0.000\,14$ |
| U-Net$_{2015}$128 | 448 | $0.026\,89 \pm 0.000\,08$ | $0.239\,41 \pm 0.001\,32$ |
| U-Net$_{2015}$128 | 5600 | $0.001\,11 \pm 0.000\,01$ | $0.012\,36 \pm 0.000\,01$ |
| U-Net$_{2015\text{-}tanh}$64 | 448 | $0.086\,29 \pm 0.003\,04$ | $0.581\,40 \pm 0.010\,45$ |
| U-Net$_{2015\text{-}tanh}$64 | 5600 | $0.005\,49 \pm 0.000\,02$ | $0.049\,60 \pm 0.000\,40$ |
| U-Net$_{2015\text{-}tanh}$128 | 448 | $0.059\,54 \pm 0.000\,36$ | $0.449\,91 \pm 0.001\,58$ |
| U-Net$_{2015\text{-}tanh}$128 | 5600 | $0.002\,65$ | $0.025\,40 \pm 0.000\,23$ |
| U-Net$_{mod}$64 | 448 | $0.021\,55 \pm 0.000\,44$ | $0.201\,22 \pm 0.004\,35$ |
| U-Net$_{mod}$64 | 5600 | $0.000\,61 \pm 0.000\,01$ | $0.010\,22 \pm 0.000\,04$ |
| U-F2Net$_{modes16,8}$ | 448 | $0.017\,78$ | $0.128\,92 \pm 0.000\,23$ |
| U-F2Net$_{modes16,8}$ | 5600 | $0.000\,36 \pm 0.000\,01$ | $0.001\,92 \pm 0.000\,02$ |

Table 7: Shallow water 1-day predictions, velocity function formulation. Rollout and one-step errors of various architectures on the shallow water equations are reported. Summed mean-squared errors (SMSE) are obtained for 1-day predictions for the velocity function formulation and are averaged over three different random seeds. If results are displayed without standard deviation, the obtained standard deviation is lower than the four digit precision minimum. The best model of each model class is highlighted.

| | | SMSE | |
|---|---|---|---|
| METHOD | Trajs. | onestep | rollout |
| ResNet128 | 256 | $0.3152 \pm 0.0039$ | $2.7775 \pm 0.0122$ |
| ResNet128 | 2800 | $0.1329 \pm 0.0010$ | $1.7711 \pm 0.0273$ |
| ResNet256 | 256 | $0.1827 \pm 0.0018$ | $1.8015 \pm 0.0037$ |
| ResNet256 | 2800 | $0.1294 \pm 0.0122$ | $1.7957 \pm 0.0947$ |
| DilResNet-128 | 256 | $0.1596 \pm 0.0097$ | $1.4477 \pm 0.0754$ |
| DilResNet-128 | 2800 | $0.0161 \pm 0.0005$ | $0.1900 \pm 0.0061$ |
| DilResNet-128-norm | 256 | $0.0749 \pm 0.0026$ | $0.6621 \pm 0.0301$ |
| DilResNet-128-norm | 2800 | $0.0102 \pm 0.0002$ | $0.1369 \pm 0.0009$ |
| FNO128-8$_{modes16}$ | 256 | $0.1561$ | $0.9394 \pm 0.0081$ |
| FNO128-8$_{modes16}$ | 2800 | $0.0122 \pm 0.0001$ | $0.0505 \pm 0.0004$ |
| UNO64 | 256 | $0.2732 \pm 0.0009$ | $2.0338 \pm 0.0071$ |
| UNO64 | 2800 | $0.0135 \pm 0.0001$ | $0.0587 \pm 0.0004$ |
| UNO128 | 256 | $0.2300 \pm 0.0039$ | $1.7308 \pm 0.0173$ |
| UNO128 | 2800 | $0.0046$ | $0.0158 \pm 0.0001$ |
| U-Net$_{base}$64 | 256 | $0.0569 \pm 0.0008$ | $0.4457 \pm 0.0038$ |
| U-Net$_{base}$64 | 2800 | $0.0043$ | $0.0316 \pm 0.0007$ |
| U-Net$_{base}$128 | 256 | $0.0428 \pm 0.0020$ | $0.3761 \pm 0.0189$ |
| U-Net$_{base}$128 | 2800 | $0.0023 \pm 0.0002$ | $0.0229 \pm 0.0022$ |
| U-Net$_{mod}$64 | 256 | $0.0272 \pm 0.0004$ | $0.2678 \pm 0.0017$ |
| U-Net$_{mod}$64 | 2800 | $0.0015 \pm 0.0002$ | $0.0243 \pm 0.0065$ |
| U-Net$_{mod}$64 | 256 | $0.0239 \pm 0.0008$ | $0.2089 \pm 0.0291$ |
| U-Net$_{mod}$64 | 2800 | $0.0014 \pm 0.0008$ | $0.0144 \pm 0.0149$ |
| U-F1Net$_{modes16}$ | 256 | $0.0257 \pm 0.0005$ | $0.1638 \pm 0.0032$ |
| U-F1Net$_{modes8}$ | 2800 | $0.0009 \pm 0.0004$ | $0.0041 \pm 0.0019$ |
| U-F2Net$_{modes16,8}$ | 256 | $0.0253 \pm 0.0016$ | $0.1710 \pm 0.0124$ |
| U-F2Net$_{modes16,8}$ | 2800 | $0.0006 \pm 0.0002$ | $0.0023 \pm 0.0009$ |
| U-F2Net$_{modes8,8}$ | 256 | $0.0248 \pm 0.0006$ | $0.1659 \pm 0.0031$ |
| U-F2Net$_{modes8,8}$ | 2800 | $0.0005$ | $0.0022 \pm 0.0003$ |
| U-F2Net$_{modes16,16}$ | 256 | $0.0315 \pm 0.0007$ | $0.2398 \pm 0.0068$ |
| U-F2Net$_{modes16,16}$ | 2800 | $0.0003$ | $0.0013$ |
| U-F2Net$_{att,modes16,8}$ | 256 | $0.0238 \pm 0.0016$ | $0.1545 \pm 0.0126$ |
| U-F2Net$_{att,modes16,8}$ | 2800 | $0.0070 \pm 0.0093$ | $0.0553 \pm 0.0841$ |

### B.4 Navier-Stokes equations.

2D Navier-Stokes data is obtained on a grid with spatial resolution of $128 \times 128$ ($\Delta x = 0.25$, $\Delta y = 0.25$), and temporal resolution of $\Delta t = 1.5\,\mathrm{s}$, a viscosity parameter of $\nu = 0.01$, and a buoyancy factor of $(0, 0.5)^T$. The equation is solved on a closed domain with Dirichlet boundary conditions ($v = 0$) for the velocity, and Neumann boundaries $\frac{\partial s}{\partial x} = 0$ for the scalar field. We run the simulation for $21\,\mathrm{s}$ and sample every $1.5\,\mathrm{s}$. Trajectories contain scalar and vector fields at 14 different time points. The inputs to the Navier-Stokes experiments are respective fields at the previous 4 timesteps. Exemplary rollout trajectories are displayed in Figure 16. We outline further details on the results in Figure 17 and Table 8. Additionally, we ablate different encoding/decoding choices for U-Net based architectures in Figures 18.

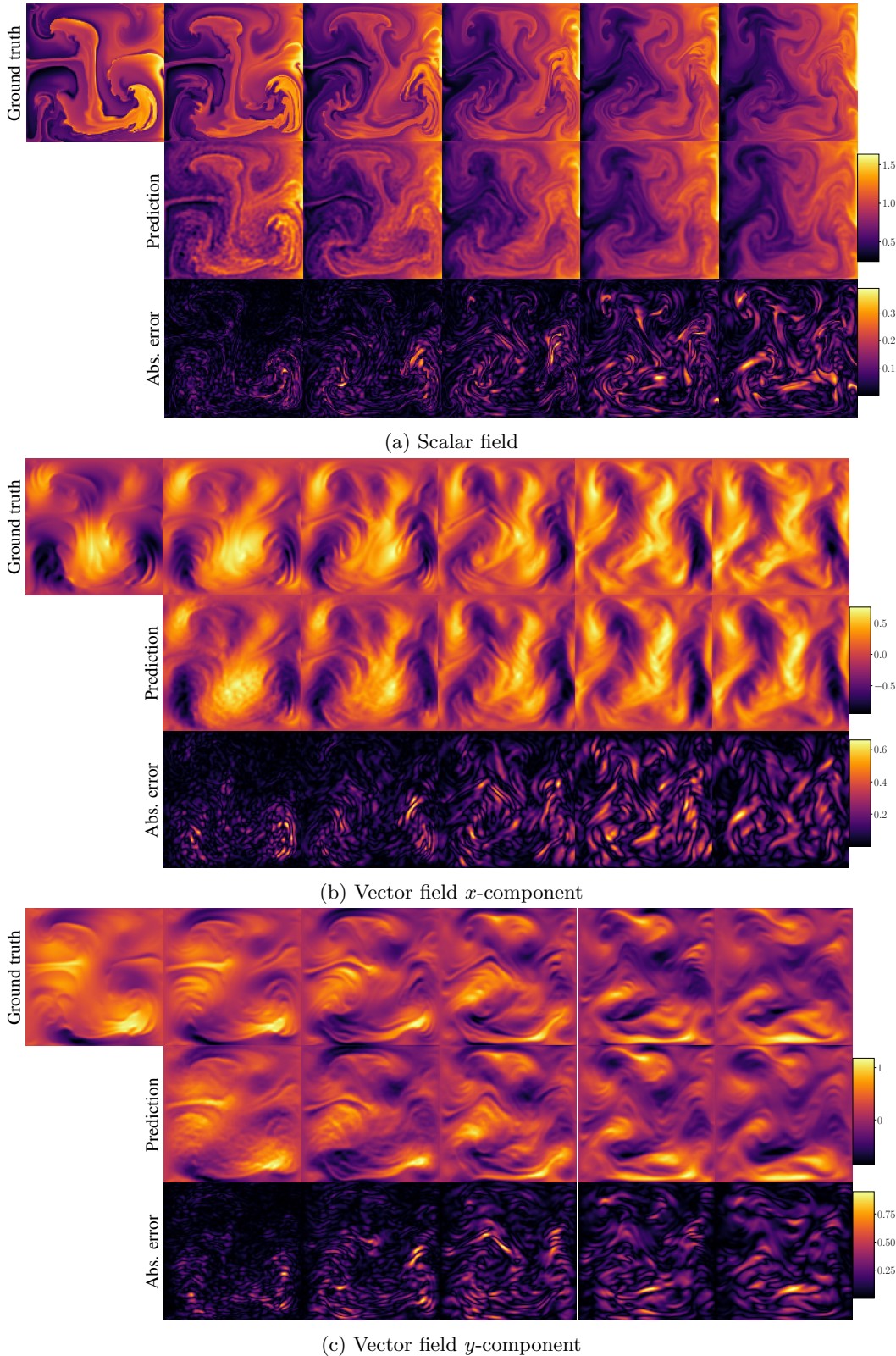

(a) Scalar field

(b) Vector field $x$-component

(c) Vector field $y$-component

Figure 15: Navier-Stokes, velocity function form. Example rollouts of the scalar and vector velocity field of the Navier-Stokes experiments are shown, obtained by a FNO96-4$_{\text{modes32,32}}$ PDE surrogate model (middle), and compared to the ground truth (top). Predictions are obtained for a time window $\Delta t = 1.5\,\text{s}$. The respective model input fields comprise four timesteps, we only show the last of those (left-most ground truth column).

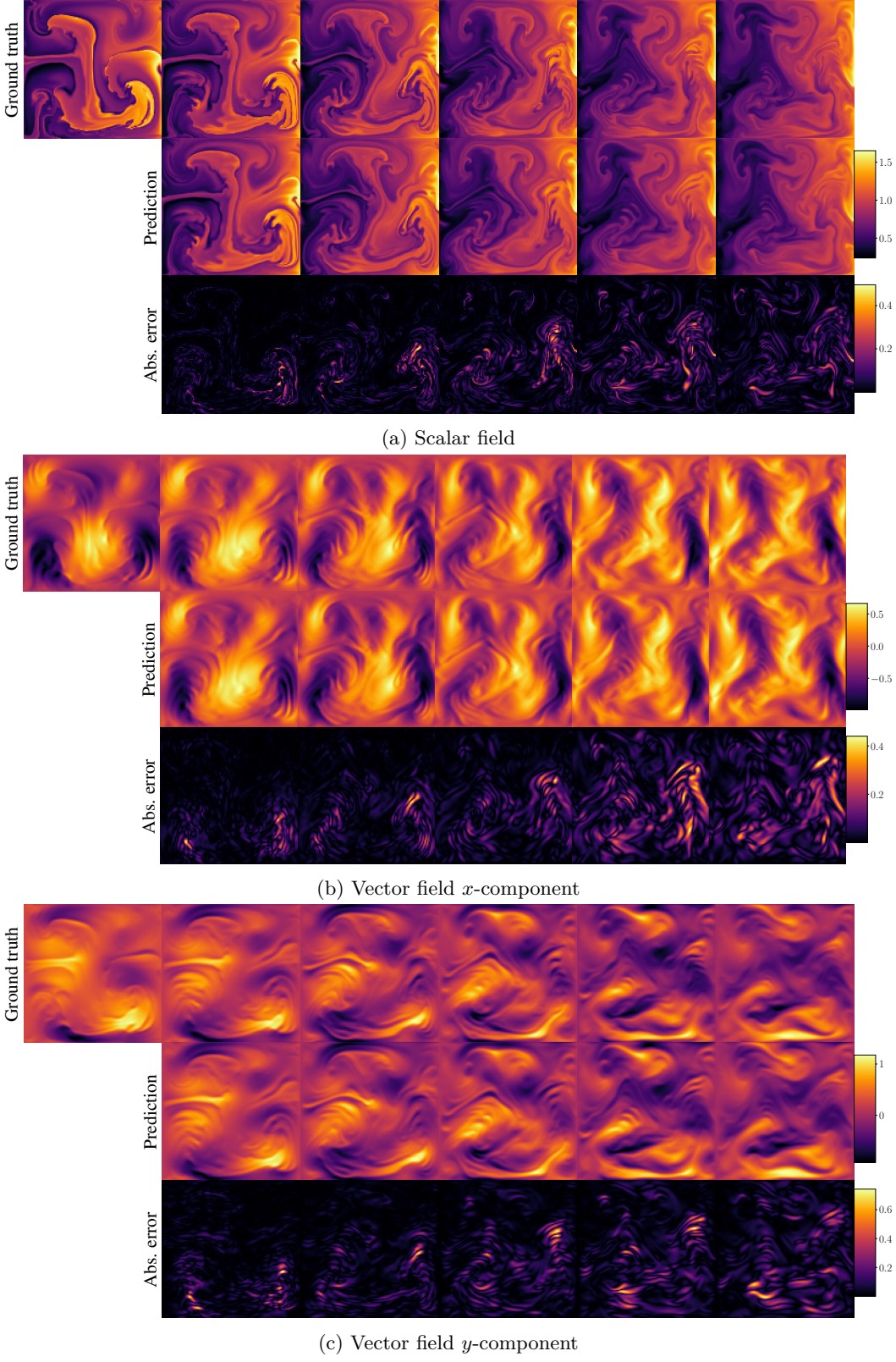

(a) Scalar field

(b) Vector field $x$-component

(c) Vector field $y$-component

Figure 16: Navier-Stokes, velocity function form. Example rollouts of the scalar and vector velocity field of the Navier-Stokes experiments are shown, obtained by a U-F1Net$_{\text{modes16}}$ PDE surrogate model (top), and compared to the ground truth (bottom). Predictions are obtained for a time window $\Delta t = 1.5\,\text{s}$. The respective model input fields comprise four timesteps, we only show the last of those (left-most ground truth column).

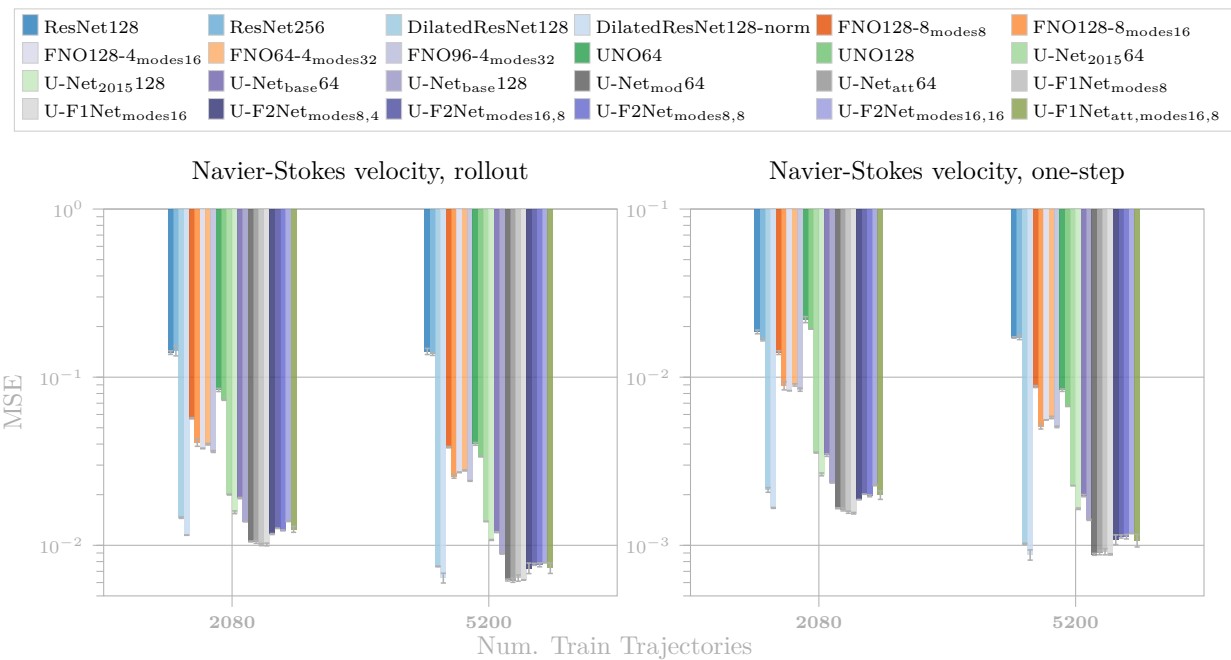

Figure 17: Navier-Stokes velocity function form. Rollout and one-step errors of various architectures on the Navier-Stokes equations are reported, obtained for predictions of 1.5 s, and are averaged over three different random seeds. Note the logarithmic scale of the $y$-axes.

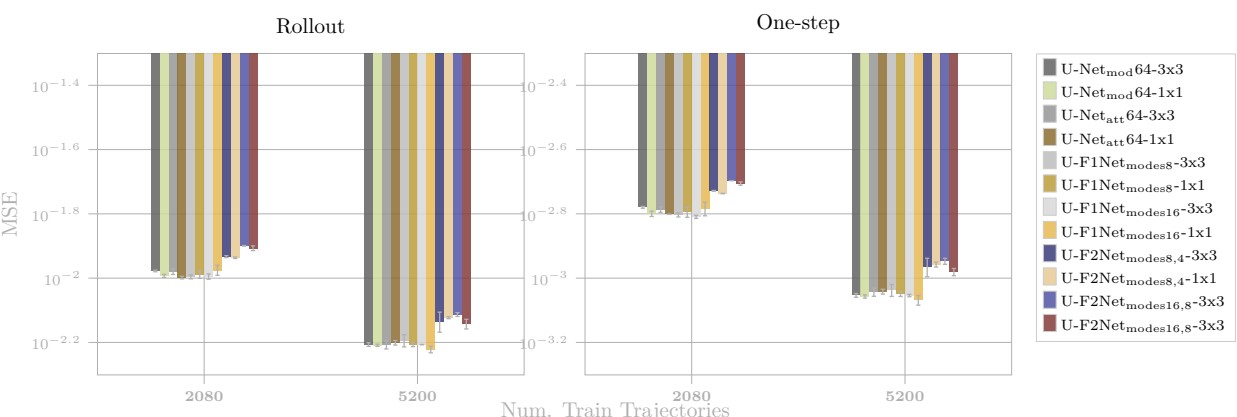

Figure 18: Navier-Stokes velocity function form. Ablation results of different encoding and decoding choices for various U-Net architectures are reported. $1 \times 1$ and $3 \times 3$ kernels are compared for both encoding and decoding. Rollout and one-step errors are obtained on the Navier-Stokes equations in velocity function form and are averaged over three different random seeds.

Table 8: Navier-Stokes, velocity function formulation. Rollout and one-step errors of various architectures on the Navier-Stokes equations are reported. Summed mean-squared errors (SMSE) are obtained and are averaged over three different random seeds. If results are displayed without standard deviation, the obtained standard deviation is lower than the five digit precision minimum. The best model of each model class is highlighted.

| | | SMSE | |
|---|---|---|---|
| METHOD | Trajs. | onestep | rollout |
| ResNet128 | 2080 | $0.01860 \pm 0.00049$ | $0.14002 \pm 0.00344$ |
| ResNet128 | 5200 | $0.01722 \pm 0.00017$ | $0.14230 \pm 0.00598$ |
| ResNet256 | 2080 | $0.01675 \pm 0.00032$ | $0.14344 \pm 0.00961$ |
| ResNet256 | 5200 | $0.01725 \pm 0.00052$ | $0.13747 \pm 0.00284$ |
| DilResNet-128 | 2080 | $0.00214 \pm 0.00007$ | $0.01460 \pm 0.00014$ |
| DilResNet-128 | 5200 | $0.00102 \pm 0.00001$ | $0.00748 \pm 0.00004$ |
| DilResNet-128-norm | 2080 | $0.00167 \pm 0.00001$ | $0.01148 \pm 0.00001$ |
| DilResNet-128-norm | 5200 | $0.00088 \pm 0.00006$ | $0.00639 \pm 0.00043$ |
| FNO128-8$_{modes8}$ | 2080 | $0.01400 \pm 0.00036$ | $0.05696 \pm 0.00051$ |
| FNO128-8$_{modes8}$ | 5200 | $0.00879 \pm 0.00012$ | $0.03836 \pm 0.00037$ |
| FNO128-8$_{modes16}$ | 2080 | $0.00890 \pm 0.00050$ | $0.04085 \pm 0.00205$ |
| FNO128-8$_{modes16}$ | 5200 | $0.00510 \pm 0.00019$ | $0.02576 \pm 0.00071$ |
| FNO128-4$_{modes16}$ | 2080 | $0.00831$ | $0.03769 \pm 0.00017$ |
| FNO128-4$_{modes16}$ | 5200 | $0.00557$ | $0.02717 \pm 0.00013$ |
| FNO64-4$_{modes32}$ | 2080 | $0.00899 \pm 0.00014$ | $0.03994 \pm 0.00045$ |
| FNO64-4$_{modes32}$ | 5200 | $0.00576 \pm 0.00009$ | $0.02787 \pm 0.00019$ |
| FNO96-4$_{modes32}$ | 2080 | $0.00843 \pm 0.00019$ | $0.03611 \pm 0.00044$ |
| FNO96-4$_{modes32}$ | 5200 | $0.00507 \pm 0.00006$ | $0.02414 \pm 0.00013$ |
| UNO64 | 2080 | $0.02200 \pm 0.00091$ | $0.08391 \pm 0.00179$ |
| UNO64 | 5200 | $0.00837 \pm 0.00014$ | $0.04010 \pm 0.00074$ |
| UNO128 | 2080 | $0.01933 \pm 0.00003$ | $0.07314 \pm 0.00035$ |
| UNO128 | 5200 | $0.00671 \pm 0.00002$ | $0.03375 \pm 0.00021$ |
| U-Net$_{2015\text{-}tanh}$64 | 2080 | $0.00651 \pm 0.00001$ | $0.03327 \pm 0.00001$ |
| U-Net$_{2015\text{-}tanh}$64 | 5200 | $0.00359 \pm 0.00001$ | $0.02051 \pm 0.00007$ |
| U-Net$_{2015\text{-}tanh}$128 | 2080 | $0.00456 \pm 0.00007$ | $0.02527 \pm 0.00043$ |
| U-Net$_{2015\text{-}tanh}$128 | 5200 | $0.00275$ | $0.01671 \pm 0.00009$ |
| U-Net$_{2015}$64 | 2080 | $0.00356 \pm 0.00002$ | $0.02004 \pm 0.00013$ |
| U-Net$_{2015}$64 | 5200 | $0.00226$ | $0.01386 \pm 0.00004$ |
| U-Net$_{2015}$128 | 2080 | $0.00264 \pm 0.00005$ | $0.01572 \pm 0.00030$ |
| U-Net$_{2015}$128 | 5200 | $0.00165 \pm 0.00002$ | $0.01076 \pm 0.00007$ |
| U-Net$_{base}$64 | 2080 | $0.00344 \pm 0.00006$ | $0.01910 \pm 0.00017$ |
| U-Net$_{base}$64 | 5200 | $0.00197 \pm 0.00003$ | $0.01197 \pm 0.00008$ |
| U-Net$_{base}$128 | 2080 | $0.00235 \pm 0.00001$ | $0.01383 \pm 0.00003$ |
| U-Net$_{base}$128 | 5200 | $0.00142 \pm 0.00001$ | $0.00898 \pm 0.00013$ |
| U-Net$_{mod}$64 | 2080 | $0.00166 \pm 0.00001$ | $0.01053 \pm 0.00006$ |
| U-Net$_{mod}$64 | 5200 | $0.00088 \pm 0.00001$ | $0.00621 \pm 0.00008$ |
| U-Net$_{att}$64 | 2080 | $0.00163 \pm 0.00003$ | $0.01048 \pm 0.00022$ |
| U-Net$_{att}$64 | 5200 | $0.00090 \pm 0.00003$ | $0.00620 \pm 0.00018$ |
| U-F1Net$_{modes8}$ | 2080 | $0.00157 \pm 0.00002$ | $0.01010 \pm 0.00015$ |
| U-F1Net$_{modes8}$ | 5200 | $0.00092 \pm 0.00004$ | $0.00639 \pm 0.00028$ |
| U-F1Net$_{modes16}$ | 2080 | $0.00155 \pm 0.00002$ | $0.01012 \pm 0.00019$ |
| U-F1Net$_{modes16}$ | 5200 | $0.00088 \pm 0.00001$ | $0.00621 \pm 0.00001$ |
| U-F2Net$_{modes8,4}$ | 2080 | $0.00187 \pm 0.00001$ | $0.01167 \pm 0.00008$ |
| U-F2Net$_{modes8,4}$ | 5200 | $0.00108 \pm 0.00007$ | $0.00732 \pm 0.00053$ |
| U-F2Net$_{modes16,8}$ | 2080 | $0.00201$ | $0.01261 \pm 0.00003$ |
| U-F2Net$_{modes16,8}$ | 5200 | $0.00113 \pm 0.00002$ | $0.00771 \pm 0.00011$ |
| U-F2Net$_{modes8,8}$ | 2080 | $0.00197 \pm 0.00003$ | $0.01227 \pm 0.00008$ |
| U-F2Net$_{modes8,8}$ | 5200 | $0.00113 \pm 0.00003$ | $0.00766 \pm 0.00020$ |
| U-F2Net$_{modes16,16}$ | 2080 | $0.00229 \pm 0.00003$ | $0.01388 \pm 0.00004$ |
| U-F2Net$_{modes16,16}$ | 5200 | $0.00118$ | $0.00790 \pm 0.00004$ |
| U-F2Net$_{att,modes16,8}$ | 2080 | $0.00199 \pm 0.00012$ | $0.01245 \pm 0.00051$ |
| U-F2Net$_{att,modes16,8}$ | 5200 | $0.00107 \pm 0.00009$ | $0.00732 \pm 0.00052$ |

Table 9: Navier-Stokes, velocity function formulation. L2 training objective of Li et al. (2020a) is used. Rollout and one-step errors of various architectures on the Navier-Stokes equations are reported. Summed mean-squared errors (SMSE) are obtained and are averaged over three different random seeds. If results are displayed without standard deviation, the obtained standard deviation is lower than the five digit precision minimum. The best model of each model class is highlighted.

| | | SMSE | |
| Method | Trajs. | onestep | rollout |
| --- | --- | --- | --- |
| DilResNet128 | 2080 | $0.00206 \pm 0.00001$ | $0.01396 \pm 0.00005$ |
| DilResNet128 | 5200 | $0.00102 \pm 0.00003$ | $0.00749 \pm 0.00012$ |
| DilResNet128-norm | 2080 | $0.00147 \pm 0.00001$ | $0.01011 \pm 0.00001$ |
| DilResNet128-norm | 5200 | $0.00082$ | $0.00604$ |
| FNO128-8$_{modes8}$ | 2080 | $0.01249$ | $0.05187$ |
| FNO128-8$_{modes8}$ | 5200 | $0.00805$ | $0.03591$ |
| FNO128-8$_{modes16}$ | 2080 | $0.00823$ | $0.03865$ |
| FNO128-8$_{modes16}$ | 5200 | $0.00484$ | $0.02485$ |
| FNO64-4$_{modes32}$ | 2080 | $0.00781 \pm 0.00014$ | $0.03594 \pm 0.00047$ |
| FNO64-4$_{modes32}$ | 5200 | $0.00517 \pm 0.00007$ | $0.02556 \pm 0.00013$ |
| FNO96-4$_{modes32}$ | 2080 | $0.00745 \pm 0.00015$ | $0.03242 \pm 0.00038$ |
| FNO96-4$_{modes32}$ | 5200 | $0.00470 \pm 0.00006$ | $0.02227 \pm 0.00011$ |
| FNO128-4$_{modes32}$ | 2080 | $0.00710 \pm 0.00004$ | $0.03080$ |
| FNO128-4$_{modes32}$ | 5200 | $0.00443$ | $0.02076$ |
| UNO64 | 2080 | $0.01707$ | $0.07089$ |
| UNO64 | 5200 | $0.00725$ | $0.03596$ |
| U-Net$_{base}$6 | 2080 | $0.00311 \pm 0.00001$ | $0.01774 \pm 0.00005$ |
| U-Net$_{base}$64 | 5200 | $0.00202 \pm 0.00001$ | $0.01222$ |
| U-Net$_{base}$128 | 2080 | $0.00212 \pm 0.00002$ | $0.01261 \pm 0.00005$ |
| U-Net$_{base}$128 | 5200 | $0.00137$ | $0.00868 \pm 0.00004$ |
| U-Net$_{2015}$64 | 2080 | $0.00324 \pm 0.00007$ | $0.01861 \pm 0.00044$ |
| U-Net$_{2015}$64 | 5200 | $0.00223 \pm 0.00001$ | $0.01376 \pm 0.00015$ |
| U-Net$_{2015}$128 | 2080 | $0.00238 \pm 0.00002$ | $0.01446 \pm 0.00002$ |
| U-Net$_{2015}$128 | 5200 | $0.00161 \pm 0.00002$ | $0.01047 \pm 0.00006$ |
| U-Net$_{mod}$64 | 2080 | $0.00151$ | $0.00975 \pm 0.00005$ |
| U-Net$_{mod}$64 | 5200 | $0.00091 \pm 0.00001$ | $0.00637 \pm 0.00002$ |

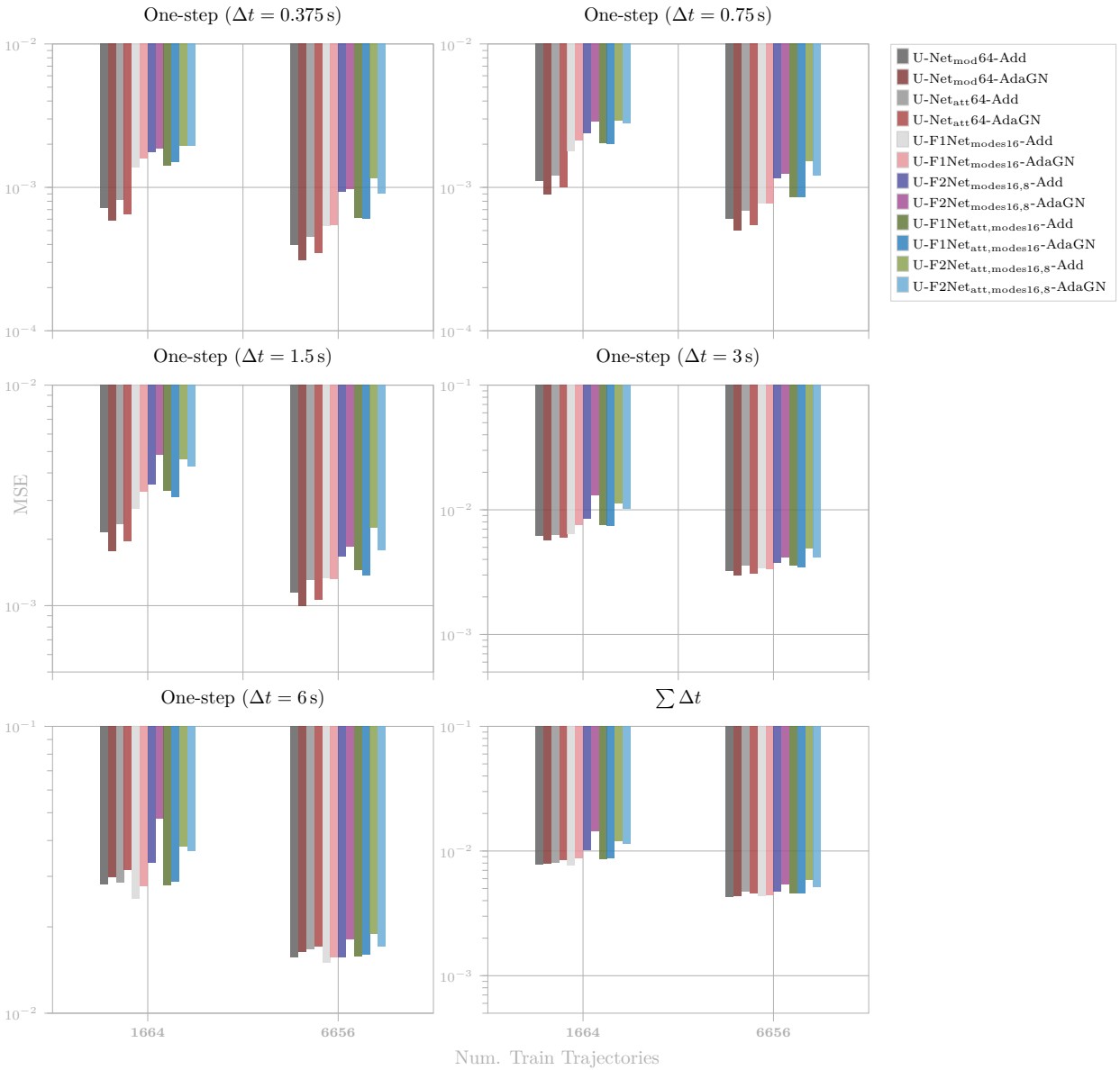

Figure 19: Navier-Stokes parameter conditioning experiments. Ablation results for different parameter conditioning methods. "Addition"(Add) and "AdaGN" are compared on one-step errors, reported for different time windows and averaged over 208 unseen values of the buoyancy force term. For low time windows, AdaGN seems to be beneficial.

### B.5 Parameter conditioning.

We use the same spatial resolutions and boundary conditions as described in Section B.4. The inputs to the Navier-Stokes parameter conditioning experiments are respective fields at the previous timestep. Exemplary rollout trajectories predicted by a single surrogate model for different buoyancy force values are displayed in Figures 20,21,22. We outline further details on the results in Table 10. Additionally, we ablate different parameter conditioning choices in Figure 19, namely "Addition" versus "AdaGN" for U-Net blocks, and "Addition" vs "Spatial-Spectral" for Fourier blocks. The default choice is "Addition".

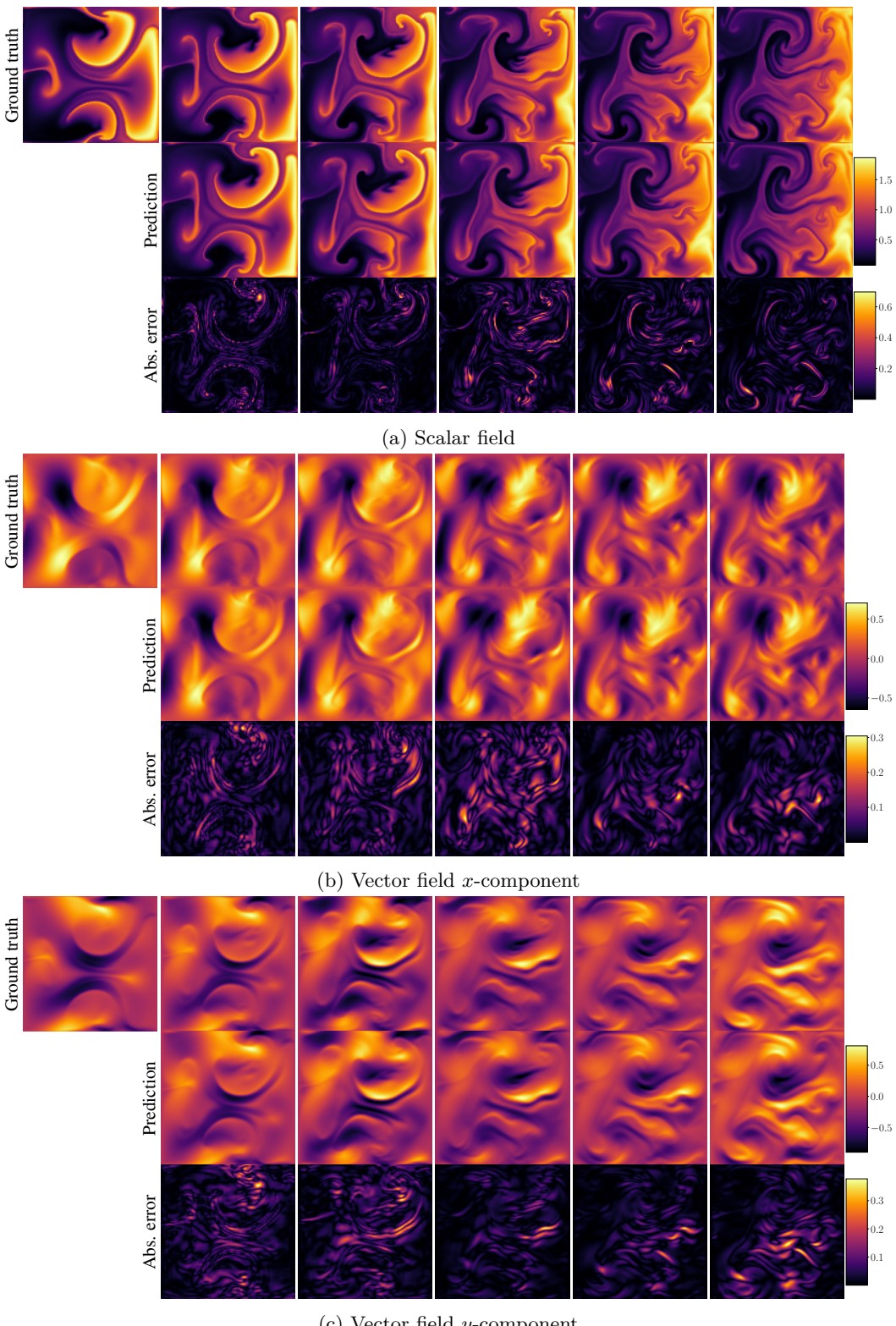

(a) Scalar field

(b) Vector field $x$-component

(c) Vector field $y$-component

Figure 20: Parameter conditioning for Navier-Stokes equations, $f = 0.21$. Example rollouts of the scalar and vector velocity field of the Navier-Stokes experiments are shown, obtained by a U-Net$_{\mathrm{mod}}$ PDE surrogate model (top), and compared to the ground truth (bottom). Predictions are obtained for a time window $\Delta t = 1.5\,\mathrm{s}$ and a buoyancy force term of $f = 0.21$. Model inputs are respective fields at the last timestep (left-most ground truth column).

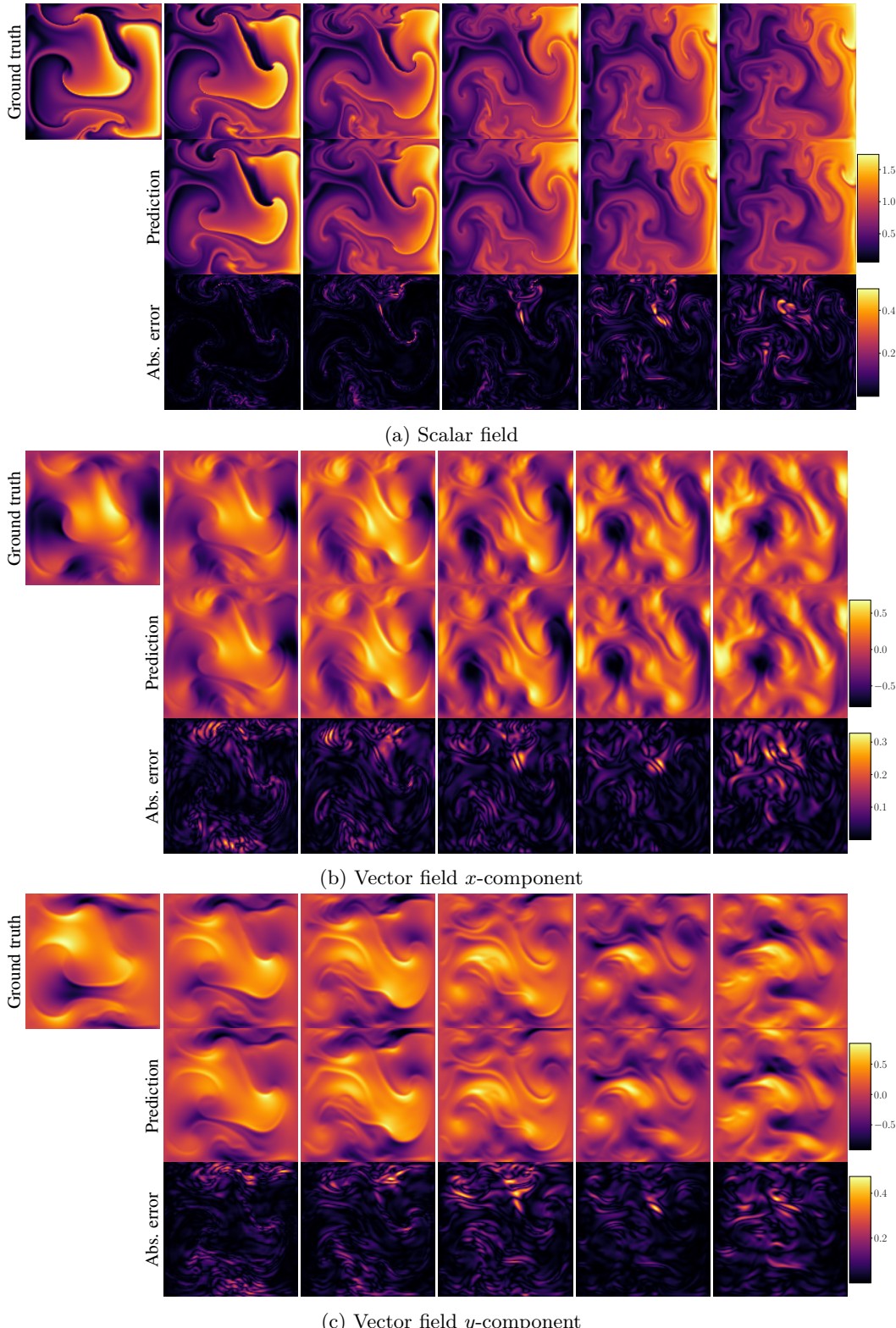

(a) Scalar field

(b) Vector field $x$-component

(c) Vector field $y$-component

Figure 21: Parameter conditioning for Navier-Stokes equations, $f = 0.33$. Example rollouts of the scalar and vector velocity field of the Navier-Stokes experiments are shown, obtained by a U-Net$_{\text{mod}}$ PDE surrogate model (top), and compared to the ground truth (bottom). Predictions are obtained at a time window $\Delta t = 1.5\,\text{s}$ and a buoyancy force term of $f = 0.33$. Model inputs are respective fields at the last timestep (left-most ground truth column).

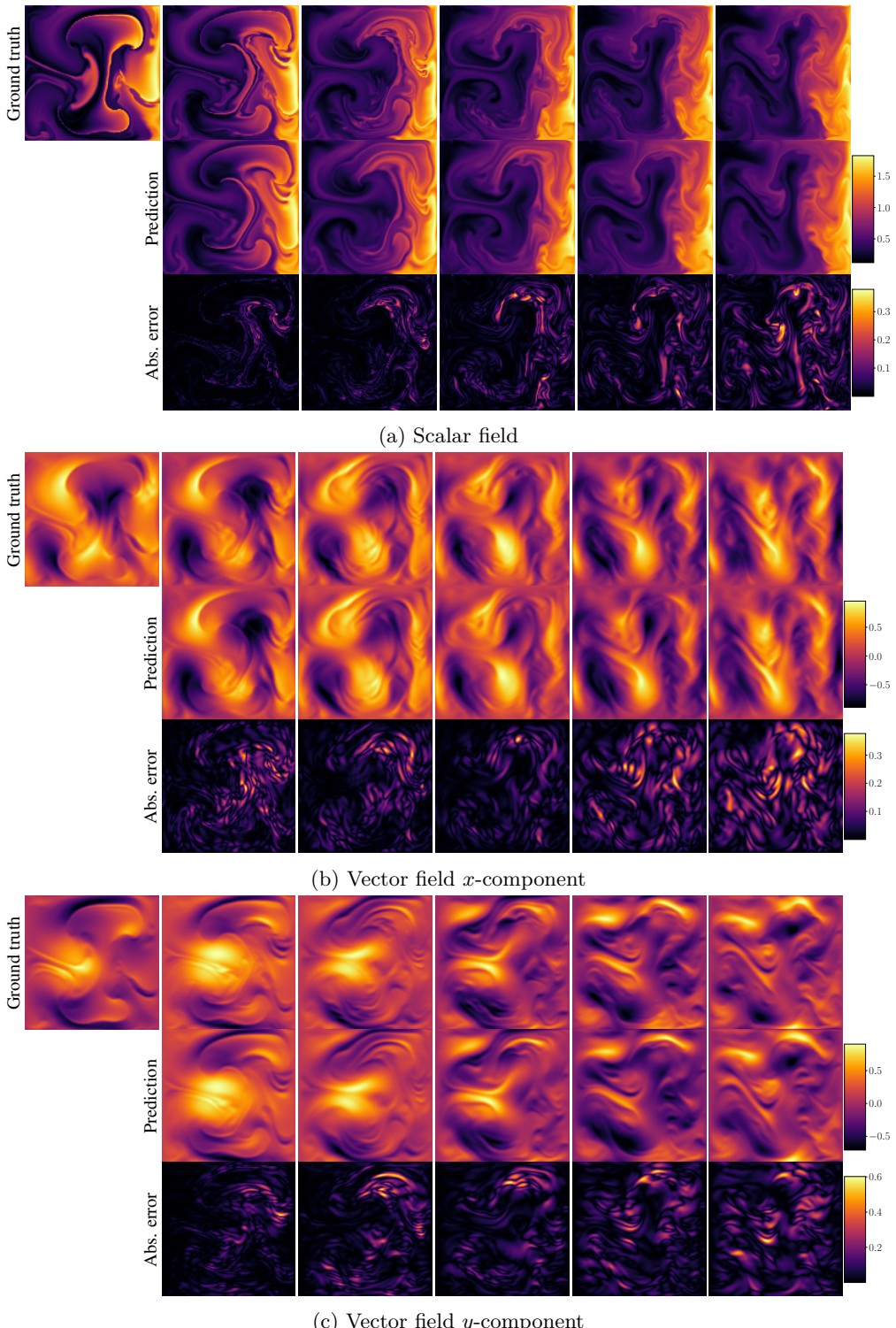

Figure 22: Parameter conditioning for Navier-Stokes equations, $f = 0.48$. Example rollouts of the scalar and vector velocity field of the Navier-Stokes experiments are shown, obtained by a U-Net$_{\text{mod}}$ PDE surrogate model (top), and compared to the ground truth (bottom). Predictions are obtained at a time window $\Delta t = 1.5\,\text{s}$ and a buoyancy force term of $f = 0.48$. Model inputs are respective fields at the last timestep (left-most ground truth column).

Table 10: Parameter conditioning on the Navier-Stokes equation, velocity function formulation. Summed mean-squared errors of various architectures are reported for different number of training trajectories, and different time windows. Conditioning results at different time windows are averaged over 208 unseen values of the buoyancy force term. The best model of each model class is highlighted. Different parameter conditioning choices are ablated, namely "Addition" versus "AdaGN" for U-Net blocks, and "Addition" vs "Spatial-Spectral" for Fourier blocks. The default choice is "Addition".

| | | SMSE | | | | |
|---:|:---:|:---:|:---:|:---:|:---:|:---:|
| METHOD | Trajs. | 0.375 s | 0.75 s | 1.5 s | 3.0 s | 6.0 s |
| $FNO128_{modes16}$ | 1664 | 0.005 17 | 0.006 93 | 0.011 73 | 0.026 66 | 0.064 23 |
| $FNO128_{modes16}$ | 6656 | 0.003 88 | 0.004 83 | 0.007 40 | 0.015 44 | 0.041 77 |
| $FNO128_{modes16}$-SpaSpec | 1664 | 0.004 62 | 0.006 67 | 0.014 04 | 0.038 34 | 0.076 48 |
| $FNO128_{modes16}$-SpaSpec | 6656 | 0.003 48 | 0.004 55 | 0.008 01 | 0.019 73 | 0.052 08 |
| $U\text{-}Net_{mod}64$ | 1664 | 0.000 72 | 0.001 11 | 0.002 16 | 0.006 22 | 0.028 05 |
| $U\text{-}Net_{mod}64$ | 6656 | 0.000 40 | 0.000 61 | 0.001 15 | 0.003 25 | 0.015 60 |
| $U\text{-}Net_{mod}64$-AdaGN | 1664 | 0.000 59 | 0.000 90 | 0.001 77 | 0.005 68 | 0.029 89 |
| $U\text{-}Net_{mod}64$-AdaGN | 6656 | 0.000 31 | 0.000 50 | 0.001 00 | 0.003 00 | 0.016 32 |
| $U\text{-}Net_{att}64$ | 1664 | 0.000 82 | 0.001 22 | 0.002 34 | 0.006 30 | 0.028 46 |
| $U\text{-}Net_{att}64$ | 6656 | 0.000 46 | 0.000 69 | 0.001 30 | 0.003 56 | 0.016 75 |
| $U\text{-}Net_{att}64$-AdaGN | 1664 | 0.000 65 | 0.001 01 | 0.001 95 | 0.005 97 | 0.031 60 |
| $U\text{-}Net_{att}64$-AdaGN | 6656 | 0.000 35 | 0.000 55 | 0.001 06 | 0.003 10 | 0.017 03 |
| $U\text{-}F1Net_{modes16}$ | 1664 | 0.001 38 | 0.001 80 | 0.002 75 | 0.006 45 | 0.024 97 |
| $U\text{-}F1Net_{modes16}$ | 6656 | 0.000 54 | 0.000 78 | 0.001 33 | 0.003 38 | 0.015 04 |
| $U\text{-}F1Net_{modes16}$-SpaSpec | 1664 | 0.000 76 | 0.001 16 | 0.002 42 | 0.007 70 | 0.034 95 |
| $U\text{-}F1Net_{modes16}$-SpaSpec | 6656 | 0.000 40 | 0.000 61 | 0.001 16 | 0.003 39 | 0.017 54 |
| $U\text{-}F1Net_{att,modes16}$ | 1664 | 0.001 42 | 0.002 05 | 0.003 33 | 0.007 62 | 0.027 95 |
| $U\text{-}F1Net_{att,modes16}$ | 6656 | 0.000 62 | 0.000 86 | 0.001 46 | 0.003 58 | 0.015 79 |
| $U\text{-}F1Net_{att,modes16}$-SpaSpec | 1664 | 0.000 95 | 0.001 51 | 0.002 74 | 0.007 67 | 0.031 93 |
| $U\text{-}F1Net_{att,modes16}$-SpaSpec | 6656 | 0.000 46 | 0.000 72 | 0.001 32 | 0.003 72 | 0.017 86 |
| $U\text{-}F1Net_{att,modes16}$-AdaGN | 1664 | 0.001 51 | 0.002 00 | 0.003 11 | 0.007 48 | 0.028 79 |
| $U\text{-}F1Net_{att,modes16}$-AdaGN | 6656 | 0.000 61 | 0.000 86 | 0.001 37 | 0.003 47 | 0.015 96 |
| $U\text{-}F1Net_{modes16}$-AdaGN | 1664 | 0.001 61 | 0.002 12 | 0.003 30 | 0.007 58 | 0.027 69 |
| $U\text{-}F1Net_{modes16}$-AdaGN | 6656 | 0.000 55 | 0.000 77 | 0.001 32 | 0.003 38 | 0.015 64 |
| $U\text{-}F2Net_{modes16,8}$ | 1664 | 0.001 78 | 0.002 40 | 0.003 57 | 0.008 50 | 0.033 46 |
| $U\text{-}F2Net_{modes16,8}$ | 6656 | 0.000 93 | 0.001 16 | 0.001 68 | 0.003 76 | 0.015 64 |
| $U\text{-}F2Net_{modes16,8}$-SpaSpec | 1664 | 0.000 71 | 0.001 18 | 0.002 33 | 0.007 26 | 0.032 87 |
| $U\text{-}F2Net_{modes16,8}$-SpaSpec | 6656 | 0.000 39 | 0.000 64 | 0.001 22 | 0.003 68 | 0.018 62 |
| $U\text{-}F2Net_{modes16,8}$-AdaGN | 1664 | 0.001 86 | 0.002 89 | 0.004 82 | 0.013 07 | 0.047 67 |
| $U\text{-}F2Net_{modes16,8}$-AdaGN | 6656 | 0.000 98 | 0.001 24 | 0.001 85 | 0.004 18 | 0.018 09 |
| $U\text{-}F2Net_{att,modes16,8}$ | 1664 | 0.001 97 | 0.002 91 | 0.004 64 | 0.011 29 | 0.038 10 |
| $U\text{-}F2Net_{att,modes16,8}$ | 6656 | 0.001 16 | 0.001 53 | 0.002 25 | 0.004 90 | 0.018 88 |
| $U\text{-}F2Net_{att,modes16,8}$-SpaSpec | 1664 | 0.000 82 | 0.001 35 | 0.002 78 | 0.008 43 | 0.034 90 |
| $U\text{-}F2Net_{att,modes16,8}$-SpaSpec | 6656 | 0.000 44 | 0.000 71 | 0.001 37 | 0.004 04 | 0.018 96 |
| $U\text{-}F2Net_{att,modes16,8}$-AdaGN | 1664 | 0.001 96 | 0.002 81 | 0.004 29 | 0.010 19 | 0.036 73 |
| $U\text{-}F2Net_{att,modes16,8}$-AdaGN | 6656 | 0.000 91 | 0.001 21 | 0.001 79 | 0.004 14 | 0.017 05 |

