# OpenReview forum: "Towards Multi-spatiotemporal-scale Generalized PDE Modeling"
_TMLR — Accepted by TMLR_

### Review · Reviewer_WYLT · 2023-02-11

**Summary Of Contributions:**

PDEs are ubiquitous in science and engineering, but solving them can be expensive. Neural surrogates have gained increasing interest over recent years for their ability to speed up computations. It is important for these networks to be able to recognize complex multi-scale spatio-temporal patterns. This paper benchmarks two kinds of approaches, namely the Fourier Neural Operator, U-Nets and ResNets. They also import recent improvements to the U-Net architecture from the generative modelling literature.

The results show that already a plain U-Net performs better than FNOs on the tested tasks. The recent improvements to U-Net further improve results. The authors show that their best methods can generalize to different PDE parameters and time scales, which is generally considered challenging.

Finally, the authors contribute their code, which should help facilitate research and benchmarking in this research area.

**Audience:**

Yes

**Broader Impact Concerns:**

No concerns specific to this paper

**Claims And Evidence:**

Yes

**Requested Changes:**

No changes requested

**Strengths And Weaknesses:**

The paper contains a rigorous experimental comparison of U-Nets and FNOs for PDE modelling. It is important for the field that such comparisons are done, because papers by authors proposing new methods are not always unbiassed and rigorous. Another strength is that the new architecture is apparently quite effective.

The paper does not contain a big new idea, but in my view not every paper needs to have one.

---

> ### Author Response · Authors · 2023-04-16
> **Response to Reviewer WYLT**
>
> We thank the reviewer for their positive comments and appreciation for our rigorous empirical exploration of ideas in PDE modeling literature. We wholeheartedly concur that unbiased comparisons are crucial, as they prevent the field from stagnating. Our findings, which frequently challenge prevailing claims in the literature, underscore the importance of this approach throughout the paper.

---

### Review · Reviewer_RNE6 · 2023-03-26

**Summary Of Contributions:**

This paper focuses on the modelling of partial differential equations (PDE) using neural network surrogates, trained to predict the next state of the PDE $t$ time steps ahead. Specifically, the paper is a experimental review of several popular baselines either from computer vision (like U-Net) or from the neural PDE modelling literature (like FNO). For this, the paper proposes a new benchmark framework for future apple-to-apple comparisons in the literature, focusing on Navier-Stokes and derivate problems. The writing is generally good, and the text is certainly clear for the most part of the paper. The structure of the paper is ok, but it could be improved at places (details below). Certainly, this is a valuable effort and one that is needed for the field of neural PDE modelling to move forward.

**Audience:**

No

**Broader Impact Concerns:**

Since this paper has the ambition to be a widely-used benchmark, I do think it makes sense to put some effort on ablations, on better structure, and clear empirical findings. To this end, while I understand that most of my suggestions would not change the paper organically (as in, the conclusions would not change), I believe they are important for the kind of paper the authors aspire to.

**Claims And Evidence:**

Yes

**Requested Changes:**

Summarizing the weakness points, the requested changes are:

- Consider extending your datasets beyond Navier-Stokes and variants. Since the paper is about 'Generalized PDE-Modelling', having 2 or even better 3 different PDEs is reasonable.

- Improve the structure of the experiments section.
  - Please, organize them in separate ablation studies in subsections.
  - Each ablation should be clearly separable and offer clear and convincing conclusions and insights. In combination, the studies should lead to the final conclusion of the paper, since this is an empirical study.
  - The experimental settings should change in a coherent and motivated fashion. Rather than changing the $\Delta t$ or other hyperparameters in an *ad hoc* fashion, use the same parameters, use a range of parameters, or motivate clearly your current use of parameters.

- Add transformers or an equivalent family of architectures, if anything, to give a fair baseline for future comparisons.

- Improve clarity in the parts of the text and the visualizations referenced in the weakness section.



**Strengths And Weaknesses:**

**Strengths**
- The contribution of the benchmark is a great one for the community, which tends to be either too specific or too broad in its comparisons.
- The clarity in general is very good, and one can readily use this paper to dive in the research field.

**Weaknesses**
- The experiments section, while quite detailed, is unstructured in my opinion, which is a pity given the appeal of the work. As a consequence, the paper currently reads as a documentation for a benchmark rather than a paper that has detailed empirical findings and insights other than some big ones (that, for instance, FNO is perhaps not as generalizable as a standard U-Net).
  - I am missing clear, structured, and systematic experiments, where 'one variable is changed at a time' such that  gradual conclusions and insights are derived, that lead to the overall conclusion. Currently, the writing of the experiments has a more free-form flow, which is in fact quite comfortable, but hard to 'map'.
  - Since this is an empirical study, I am missing clear ablations that highlight specific aspects of the comparisons, the baselines, the data. For instance, do we need more or fewer layers with PDE-modelling? Is this somehow related to the complexity of the PDE, how chaotic it is (eg in terms of Raynolds number)? Do we need bigger or smaller latent features for U-Nets? Is regularization necessary, since the domain appearance changes over time? Are frequency modelling with FNO blocks more important than depth in Figure 4? (and many more questions like that) Currently, there are experiments like that, but they are not clearly organized. The experiment regarding conditioning on parameters is in a good direction.
  - Not all empirical findings are sufficiently explained. Why are dilated convolutions work so much better for Navier-Stokes than shallow water (compared to other methods)? Perhaps Navier-Stokes requires larger 'context', for which convolutional filters of varying size are important. That said, and considering the claim that dilated convolutions are a good next step, in my experience in computer vision, this type of adaptability does not really pay of all that much (although it can be helpful). Rather, even having multiple filters per layer of different sizes/scales is good enough, simpler to implement, and likely more sample efficient (like in Inception architectures). Perhaps it makes sense to include a baseline with multi-size filters as well.
  - Since this is an empirical study using state-of-the-art computer vision architectures, it is rather odd to leave transformers out. Truth be told, transformers are mentioned in the limitations of the paper. However, given its claims, the paper will not be complete without a transformer baseline, even a simple one.
  -  Since the paper is about 'Generalized PDE-Modelling', I believe that using only a singe 'PDE dataset' and variants, is not sufficient. Surely, Navier-Stokes is the standard PDE to solve but I believe it is not enough to claim generality. Put otherwise, how certain are the authors that the reported findings would generalize to different PDEs? And can a third reader realistically derive these type of insights from just one dataset.
- The clarity at places should be improved. Some examples in (almost) page order.
  - 'Local and global information is therefore treated in a more distributed fashion than in FNO like architectures.' $\rightarrow$ what does distributed mean here? I suppose spatial distribution (because one can also think of frequency distribution)?
  - 'Common practice is to use rather small filters' $\rightarrow$ This is common practice in computer vision, the idea being that with smaller filters one can build deeper non-linear networks, and the cascade of filters anyways leads to implicitly larger receptive fields. Is this the same for PDEs, and if yes why? I can also imagine that in PDEs the low-level patterns and textures are simpler compared to natural images, thus larger filters might be sufficient (in that shallower networks would be good enough).
  - 'We hypothesize that replacing convolutions with FNO layers which set Fourier modes above cut-off frequencies to zero might be advantageous, especially in the lower parts of the downsampling blocks of U-Net architectures' $\rightarrow$ While intuitively I think I understand what the authors mean (that near the bottleneck layers the learned features reflect more global/high-level patters, for which likely higher-frequencies associated with low-level textural patterns are not important), it is good if the paper motivates the hypothesis.
  - Can you explain why you include the 'scalar vorticity stream function form' for the Navier-Stokes? It might be used more in the PDE world, however, it is not clear what is the difference from a machine learning perspective. Does it lead to different types of solutions for the PDE?
  - Can you clarify what do you mean here 'The chosen data sets to do so consist of solution pairs u,u′ ∈ U where the pair itself is from the same solution space U , but different pairs {u,u′}1 and {u,u′}2 are from different solution spaces U1 and U2 characterized by different force terms.' Intuitively, I understand what you want to say, but I find the text unnecessarily complex.
  - End of page 7: 'ater' $\rightarrow$ 'water'
  - 'In general, we observe that conditioning is more difficult for FNO layers, most strikingly seen in the performance curves of FNOs.' $\rightarrow$ Isn't this sort of expected, since one combines information from different domains/geometries. FNO encodes frequency information, while the conditioning variable would (I suppose) be better seen as the spatial/time domain.
 - Colors in the bar charts are not easy to distinguish.
 - By rollout and one-step loss, do you use the true next state of the PDE when making the prediction (like in curriculum learning, only for the test time unrolling here)? Or you assume no access to the background simulator and the model relies only on its own predictions? If the latter is the case, I would expect an ablation also on the error accumulation and the parameters that are influenced by (eg the type of PDE etc). If the former is the case, why is this setting chosen and is it realistic to expect to have a simulator running in the background? It would still make sense when examining real observations (not simulated data), however, the study focuses on synthetic simulated data.

---

> ### Author Response · Authors · 2023-04-16
> **Response to Reviewer RNE6 (Part 1)**
>
> We are grateful to the reviewer for their detailed comments and suggestions.
>
> **missing clear, structured, and systematic experiments, where 'one variable is changed at a time'**
>
> The general idea is to make the tasks harder and harder, and systematically ablate design choices of most recent neural PDE surrogates and variants thereof on those. We therefore go from shallow water vorticity stream, shallow water velocity function to Navier-Stokes velocity function with higher spatial resolution, and finally to generalization of Navier-Stokes equation w.r.t to time scale and equation parameters. We now clarified this structure in the paper, i.e.,
>
> (i) we have added paragraphs to explain differences between vorticity stream vs velocity function formulation, and why such a comparison is important for neural PDE surrogates;
>
> (ii) we have added another paragraph that explains our motivation for choosing this experiment structure;
>
> (iii) we finally now outline our findings much clearer in the conclusion.
>
> **missing clear ablations that highlight specific aspects of the comparisons, the baselines, the data**
>
> The scope of this paper is to test many different architecture families, each of which have individually been fine-tuned on a diverse set of tasks. Such a comparison is systematically lacking in literature. For each family, we report a lot of different design choices, together with runtime and memory consumptions. This should help the practitioner on the decision of the corresponding PDE surrogate. Questions like how many layers we need for a given complexity of a PDE, or what the ideal latent space is for UNet architectures are extremely difficult to answer. In general, there is no formula for network design in deep learning, which is also true for neural PDE modeling. If the reviewer is aware of any papers in that direction, we are happy to include those. We further remark that our UNet architectures are taken from state-of-the-art computer vision models, our FNO models are significantly improved upon the FNO models reported in e.g. Li et al., and our Dilated ResNet architectures improve upon the architectures reported in Stachenfeld et al.. Finally, since we have open-sourced our codebase, the practitioner can easily run a hyper-parameter sweep on any of those.
>
> **Why are dilated convolutions work so much better for Navier-Stokes than shallow water (compared to other methods)?**
>
> To be honest we don’t know for sure either. We can make some educated speculations w.r.t. importance of aliasing vs. patterns in the data. Concretely, for data with different proportions of higher frequency modes, the importance of aliasing might shift. We have added our interpretation to the paper. Finally, we want to add that Dilated Convolutions are _much_ slower though in wall-clock time as highlighted in Table 1.
>
> **multi-size filters baseline**
>
> The main reason why we included Dilated ResNets is that they were advertised in Stachenfeld et al. (2021) and seem to perform better than UNets and FNOs. We decided to stick to the proposed architecture but improve their performance by increasing channels and optimizing normalization schemes. Again, one claim of our paper is that state-of-the-art UNets from computer vision should be considered as number one baseline for convolution based architectures, which again holds in this comparison.
>
> **the paper will not be complete without a transformer baseline, even a simple one.**
>
> While more comparisons and baselines are always good, they also come with their own requirements for compute time. Moreover, the scope of this work was limited to architectures based on convolution based primitives. We expect ViTs to work quite well given our experience on different projects, but they require 1) a lot more data than provided for these experiments, and/or 2) extra regularization tricks to not overfit to the data resulting in an overall high compute requirements. Unfortunately we no longer have access to such compute to perform decent and fair optimization at the moment. Hopefully the community can build upon our open-source project to continue these comparisons. We have open-sourced data generation and training code to make such comparisons.
>
> **meaning of distributed in the context of “Local and global information is therefore treated in a more distributed fashion than in FNO like architectures.”**
>
> Here distributed refers to the representation being “distributed” over the entire architecture i.e. multiple layers specialize into different contexts. This is in contrast to FNOs where each layer processes both local and global information.

---

> > ### Author Response · Authors · 2023-04-16
> > **Response to Reviewer RNE6 (Part 2)**
> >
> > **it is good if the paper motivates the hypothesis (of the utility of FNO in the lower parts of the downsampling blocks of U-Net)**
> >
> > We write “Bandlimited pre-subsampling (Mallat, 1999), i.e. suppressing high-frequencies before down-sampling, is a well know technique in signal processing; for an illustrative example see e.g. Figure 1 in Worrall & Welling (2019). We hypothesize that replacing convolutions with FNO layers which set Fourier modes above cut-off frequencies to zero might be advantageous, especially in the lower parts of the downsampling blocks of U-Net architectures.” Due to the Nyquist sampling theorem higher frequencies are more important in the first two downsampling blocks. We have added two sentences to make our intuition clearer.
> >
> > **only one PDE dataset and its variants are used**
> >
> > We agree with the reviewer that dataset choices are Navier-Stokes centric. However, as the governing equations in fluid dynamics, Navier-Stokes equations offer many different aspects on which neural PDE surrogates can be ablated. We are therefore confident enough that the problem sets are rich enough to get an understanding of the different model performances. Concretely, we test on two prominent variants, we test for vorticity and velocity stream function formulation, and finally we test for generalization of time and equation parameters. For all these tasks we test for up to 40 different fine-tuned architectures and different training set sizes. However, we are happy to change the title of the paper if the reviewer finds for example “Large-scale study of neural PDE surrogates on fluid mechanics problems” more appropriate.
> >
> > **why you include the 'scalar vorticity stream function form' for the Navier-Stokes?**
> >
> > We include the scalar vorticity stream function for the shallow water equation as commonly done in past work in this domain, see e.g. Li et. al. (2020). In practical problems, the velocity function formulation is more common, and thus we were ourselves curious whether we see performance differences. Especially,  since velocity fields are more intuitive from the perspective of actual measurements. We have extended the paragraph to explain in more detail why stress-testing neural PDE surrogates on both formulations is important in our opinion.
> >
> > **Can you clarify what do you mean here 'The chosen data sets to do so consist of solution pairs u,u′ ∈ U where the pair itself is from the same solution space U , but different pairs {u,u′}1 and {u,u′}2 are from different solution spaces U1 and U2 characterized by different force terms.'**
> >
> > We mean that input and target for the model are from the same solution space, i.e., have the same forcing terms, but for different data points (different input-target pairs), different forcing terms occur. We clarified this in the reworked version. We have added “Simply put, an input-output pair is always drawn from one trajectory and therefore naturally shares the same PDE parameters”
> >
> > **Isn't this sort of expected, since one combines information from different domains/geometries. FNO encodes frequency information, while the conditioning variable would (I suppose) be better seen as the spatial/time domain.**
> >
> > We describe our “spatial-spectral” parameter conditioning in Appendix B.2.6. Parallel to us https://openreview.net/forum?id=B-dM7df9Axo explored similar ideas about how to do conditioning with FNO layers. Even if obvious, we believe it was worth doing our best to make a fair comparison under practically useful parameter conditioning scenarios.
> >
> > **colors**
> >
> > Given the variety of architectures compared we tried our best to come up with a consistent color scheme:
> > ```python
> > import matplotlib as mpl
> > tab40_colors =  mpl.colors.ListedColormap(np.vstack([mpl.cm.tab20c.colors, mpl.cm.tab20b.colors]))
> > ```
> > Hopefully inclusion of actual numbers in the tables is more clear. Happy to redraw figures in case you have alternative recommendations for a color scheme.
> >
> >
> > **access to the background simulator**
> >
> > During rollouts we assume no access to the background simulator because as the reviewer points out it wouldn’t be realistic. Truly long horizon simulations are still an active area of research, however as we show here, for less than 10 steps of rollout, 1-step error and multi-step rollout errors are quite correlated. Moreover, time-conditioning helps to side-step some of the issues with auto-regressive error accumulation. These results are consistent across all methods and different PDEs under our study. We hope to show progress on this particular problem of very long horizon rollouts in future work.
> >
> > **References:**
> >
> > Li et al. Fourier neural operator for parametric partial differential equations. ICLR 2021
> >
> > Stachenfeld et al. Learned coarse models for efficient turbulence simulation. ICLR 2022

---

### Review · Reviewer_rik3 · 2023-04-04

**Summary Of Contributions:**

This submission makes a conceptual comparison between Fourier transform and convolution. It also conducted a series of experiments to compare learning models including Fourier Neural Operators (FNO), U-Net, ResNet, and Fourier U-Net.

**Audience:**

No

**Claims And Evidence:**

No

**Requested Changes:**

If this submission aims to improve over previous models, the submission needs to include sufficient innovations. If this paper plan to provide a benchmark of different methods, then it needs to include more methods and provide better analysis of different methods.

**Strengths And Weaknesses:**

This submission contains mostly introductory content, and I cannot see much innovations. Section 2 is an introductory section. The content before 3.1 introduces three architectures. The only new architecture is Fourier U-Net. Section 3.1 is also like an introductory section.

The experiment section does not seem to give definite conclusions. While this section provide rich experiment results, there is not much insights. For example, what can be reasons for different performances.

One question: has any of these benchmark problems been used by previous work? If so, then it is nice to include results from previous work.

---

> ### Author Response · Authors · 2023-04-16
> **Response to Reviewer rik3**
>
> We thank the reviewer for their comments. While comparisons against even more methods would definitely be great, that would require even more computational resources than what we have access to right now. However, we disagree that the current set of comparisons aren’t substantial or significant.
> 1. No other work has attempted to independently analyze such a variety of operator learning approaches on expert written PDE solver data. Pushing the state-of-the-art requires better understanding of the current state of approaches.
> 2. Despite the dependence of the actual utility of PDE surrogates on their ability to generalize to differently parametrized PDEs most works in this space don’t go beyond trying to generalize to different initial conditions. We believe our work presents a useful step-forward towards such an analysis.
> 3. We provide a well documented modular codebase following current best practices in scalable deep learning which can be easily extended to new data sources and modeling techniques.
>
> **has any of these benchmark problems been used by previous work? If so, then it is nice to include results from previous work.**
>
> Versions of the benchmark problems have been used in Brandstetter et al. (2023), and in Takamoto et al. (2022). We wish to note that unlike the current practice of writing naive PDE solvers to generate data to benchmark against, it’s more appropriate to leverage the expertise of experienced PDE domain experts and use their solvers to generate data to benchmark against. It also avoids the problem of overfitting the data generation and data fitting approaches to each other.
>
> **it needs to include more methods and provide better analysis of different methods.**
>
> We believe that the currently included methods are quite comprehensive and sufficiently cover the approaches using some sort of convolutional kernel in their architecture and as such provide a fair amount of insight for any practitioner in this domain:
> 1. Modern U-Nets used in computer vision (which have been optimized quite a bit for different hardware too, thanks to the heavy usage the image/video generation communities) are actually great for many of the PDE surrogate tasks for which new architectures like “Fourier Neural Operator” have been proposed but haven’t been properly compared. For example, other recently introduced benchmarks used `tanh` activations for UNets vs `GeLU` for FNOs (Takamoto et al. (2022)).
> 2. Time conditioning can be a powerful approach to getting around some of the issues of long horizon rollouts.
> 3. The utility of PDE surrogates in practice, depends on their ability to generalize to new conditions which should include different equation parameters and not just initial conditions. Again few clear comparison benchmarks have been done for different approaches under this condition. We again find that modern U-Nets do quite well compared to FNOs.
>
> Understanding these empirical results was critical for us to come up with a way to combine key ideas behind FNOs, i.e. dropping higher frequency modes, with modern U-Nets. It’s more useful in the early layers rather than later on to improve on their success.
>
> **the submission needs to include sufficient innovations.**
>
> Moreover, while we disagree that every paper needs to come up with a new architecture for it to be significant, such an expectation also runs counter to [TMLR guidelines](https://jmlr.csail.mit.edu/tmlr/reviewer-guide.html#:~:text=A%20review%20should%20have%20the%20following%20content.%20Summary,that%20you%20think%20require%20attention%20from%20the%20authors.) which state:
>
>
> > Crucially, it should not be used as a reason to reject work that isn't considered “significant” or “impactful” because it isn't achieving a new state-of-the-art on some benchmark. Nor should it form the basis for rejecting work on a method considered not “novel enough”, as novelty of the studied method is not a necessary criteria for acceptance. We explicitly avoid these terms (“significant”, “impactful”, “novel”), and focus instead on the notion of “interest”. If the authors make it clear that there is something to be learned by some researchers in their area from their work, then the criteria of interest is considered satisfied.
>
> **References**:
>
> Takamoto et al. PDEBench: An extensive benchmark for scientific machine learning. Advances in Neural Information Processing Systems 2022
>
> Brandstetter et al. [2209.04934] Clifford Neural Layers for PDE Modeling. ICLR 2023

---

### Author Response · Authors · 2023-07-07
**Gentle reminder**

We kindly ask the involved parties to address our reviews and our updated paper. It is under review for more than 6 months now.

---

### Decision · Action_Editors · 2023-07-01

**Recommendation:** Accept with minor revision

**Comment:**

The reviews were somewhat lukewarm, with the authors acknowledging that there is no single "big idea" in the paper - the main contribution is broad experimental comparison of reasonable baselines and documentation of various empirical observations.  Experiments fall short of providing a deeper understanding of architecture and data-complexity as a function of dynamics complexity (e.g., if a system is near chaotic, do we need deeper/larger architectures and larger datasets); the analysis is restricted to a only a few experimental settings; and wildly popular architectures like Transformers are missing from the analysis.

As the same time, TMLR evaluation criteria emphasizes accurate+clear evidence of claims, together with relevance to some TMLR audience. The paper satisfies these criteria, within the narrow scope of the claims it makes.

Minor revisions: Please address the final reviewer remarks on lack of connections with previous work, with more comprehensive coverage in related works. Consider reporting a transformer baseline for completeness, though this is not a requirement.


**Audience:**

The scientific modeling community is increasingly adopting deep learning based models of physical phenomenon, often combining these with domain-specific priors. Which architectures work well, how far into the future can one reliably predict, which integrators to use etc all add up to a complex design space. As such, some subset of the "ML for science" community is certainly expected to turn towards TMLR for some partial answers that are best addressed by releasing a benchmark for PDE modeling techniques and documenting baselines, which is the primary contribution of this work.

**Claims And Evidence:**

This is largely an empirical paper that compares several architectural choices for modeling fluid mechanics phenomena, which can be an incredibly challenging class of problems due turbulence and other effects arising at different time-scales. The primary contribution of the paper is a comprehensive experimental comparison between U-Net, FNO and ResNet architectures; proposing a new blended architecture that performs well, and an open source benchmark for PDE modeling.

The paper may provide useful documentation of various observations along the lines of compute and memory requirements vs error tradeoffs for different architectures; UNet variants constantly outperforming FNO; that comparisons can be sensitive to different parameterization of Naivier-Stokes equations; that it is feasible to train neural net surrogates both across initial conditions and equation parameters, etc. This documentation is clearly presented with a large appendix providing many details.

---

> ### Author Response · Authors · 2023-07-19
> **Camera-ready version uploaded**
>
> We have uploaded the camera-ready version with all reviewer comments worked in. Specifically, we made gave more background on the used partial differential equations, a more detailed description of the tested methods, referenced the most recent relevant papers in fluid mechanics and adjacent fields and open-sourced our codebase. The official codebase is now linked in the paper.